# Plan Then Action: High-Level Planning Guidance Reinforcement Learning for LLM Reasoning

**Zhihao Dou** [* 1] **Qinjian Zhao** [* 2] **Zhongwei Wan** [* 3] **Dinggen Zhang** [2] **Weida Wang** [4] **Benteng Chen** [5]
**Towsif Raiyan** [1] **Qingtao Pan** [1] **Yang Ouyang** [6] **Chaoda Song** [1] **Zhiqiang Gao** [† 2 7] **Shufei Zhang** [† 8]
**Sumon Biswas** [1]

## Abstract

Large language models (LLMs) demonstrate strong reasoning abilities via Chain-of-Thought (CoT), but their token-level generation encourages local decisions and lacks global planning, often leading to redundant or inaccurate reasoning. Existing methods, such as tree-based search and reinforcement learning (RL), attempt to address this issue but incur high computational costs and still struggle to produce reliable reasoning trajectories. To address these challenges, we propose **P**lan-**T**hen-**A**ction Enhanced Reasoning with **G**roup **R**elative **P**olicy **O**ptimization (**PTA-GRPO**), a two-stage framework designed to jointly improve high-level planning and fine-grained CoT reasoning. Specifically, in the first stage, a given LLM is responsible for summarizing CoT reasoning into compact high-level guidance, which is then leveraged for supervised fine-tuning. Then, we introduce a guidance-aware reinforcement learning method that jointly optimizes the final output and the quality of guidance, enhancing reasoning effectiveness. We evaluate PTA-GRPO on ten reasoning benchmarks across mathematics and natural sciences, using five diverse base models spanning multiple data modalities. The results show that PTA-GRPO consistently delivers significant improvements across models and tasks, demonstrating strong effectiveness and generalization.

---

[*]Equal contribution , [†]Corresponding authors. [1]Case Western Reserve University, Cleveland, OH, USA [2]Kean University, Union, NJ, USA [3]The Ohio State University, Columbus, OH, USA [4]Fudan University, Shanghai, China [5]The University of Hong Kong, Hong Kong, China [6]North Carolina State University, Raleigh, NC, USA [7]Wenzhou-Kean University, Wenzhou, China [8]Shanghai Artificial Intelligence Laboratory, Shanghai, China. Correspondence to: Zhiqiang Gao <zgao@wku.edu.cn>, Shufei Zhang <zhang-shufei@pjlab.org.cn>.

*Proceedings of the $43^{rd}$ International Conference on Machine Learning*, Seoul, South Korea. PMLR 306, 2026. Copyright 2026 by the author(s).

## 1. Introduction

Large Language Models (LLMs) have recently demonstrated remarkable reasoning abilities across a wide range of complex tasks (Xu et al., 2025a; Plaat et al., 2024; Ke et al., 2025), such as mathematics (Zhang et al., 2024; Wu et al., 2024a; Liu et al., 2023) and programming (Jiang et al., 2024), by leveraging Chain-of-Thought (CoT) reasoning (Wei et al., 2022). Models with strong reasoning capabilities, including Qwen-3 (Yang et al., 2025), DeepSeek-R1 (Wu et al., 2024b), Seed-1.5 thinking (Seed et al., 2025), and GPT-5 thinking (OpenAI, 2025), adopt CoT as a central mechanism to structure their reasoning processes. However, CoT decoding in LLMs is still a token-level Markov Decision Process (MDP) (Ouyang et al., 2022; Wan et al., 2025a; Liu et al., 2025): the output of each token is determined by the context sequence generated previously. Under this setting, mainstream decoding is both autoregressive (each decision conditions only on the prefix) and locally greedy (it optimizes short-horizon token likelihood, e.g., via greedy/low-temperature choices). This combination preserves local consistency but offers little global planning, often yielding redundant or drifting chains of thought and propagating early mistakes across long horizons (Yao et al., 2023; Qu et al., 2025; Wan et al., 2025a).

Prior work augments LLM reasoning with tree-style algorithms (Zhang et al., 2024; Yao et al., 2023; Wang et al., 2024a) such as Monte Carlo Tree Search (Zhang et al., 2024) or heuristic generation tree (Li et al., 2025) to widen exploration beyond single-path decoding. While effective in some cases, these approaches hinge on repeated external queries to the LLM, incurring substantial time and compute (Wang et al., 2024a). Crucially, they do not strengthen the model's internal reasoning: performance stems from outside search. When the model cannot verify intermediate steps, the search simply amplifies bad branches and collapses (Feng et al., 2023). In parallel, recent works inject reflection or backtracking behaviors via RL (Wan et al., 2025a; Wang et al., 2025; Gandhi et al., 2025). Such behaviors can, in principle, re-route trajectories and escape local optima (Gandhi et al., 2025; Liang et al., 2026a). Yet when triggered on corrupted

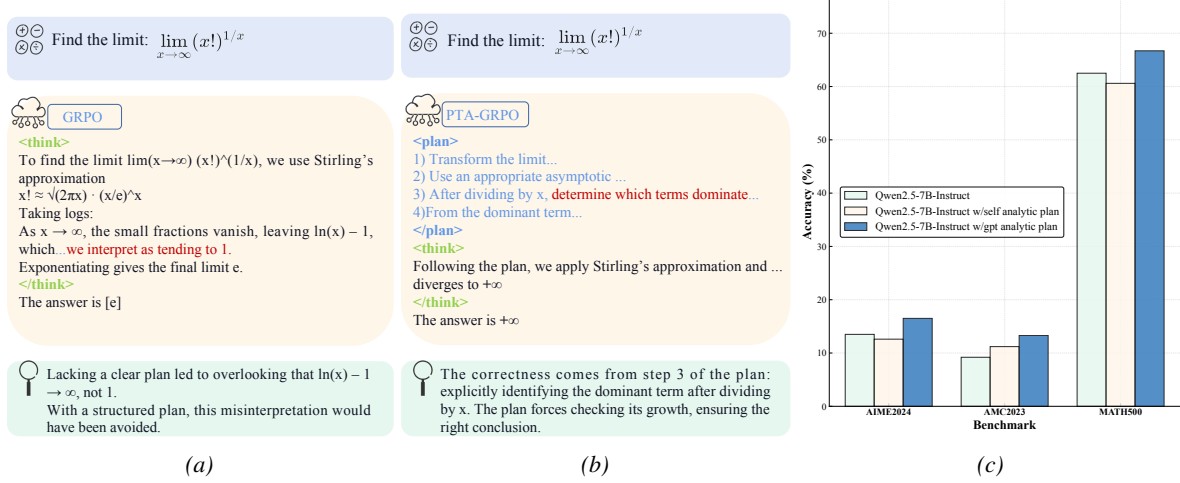

*Figure 1.* (a) GRPO reasoning process. (b) PTA-GRPO reasoning process. (c) Impact of analytic plan. In (c), the accuracy of different reasoning modes, where Qwen2.5-7B-Instruct is considered as the base model. Green indicates the base model using CoT reasoning, yellow indicates the base model reasoning with its own self-generated analytic plan, and blue indicates the base model reasoning with an analytic plan generated by GPT-o1. More test cases of PTA-GRPO are shown in Appendix A.11.

partial solutions, the model tends to reflect on its own errors, reinforcing them and drifting farther from the correct path. This occurs largely due to the absence of a global plan to guide self-reflection, leaving the model without a reliable mechanism to recover. These limitations motivate a new paradigm that *improves internal planning rather than relying on external search or post-hoc self-correction*.

Motivated by human problem-solving behavior, where an abstract plan is sketched before execution (Kahneman, 2011), we explore whether LLM reasoning can similarly benefit from an explicit plan-then-execute paradigm. Concretely, an LLM first generates a compact analytic plan that provides global guidance—such as subgoal decomposition and execution order—and then conditions detailed chain-of-thought (CoT) reasoning on this plan. This global conditioning mitigates local myopia and reduces redundant or inconsistent reasoning trajectories. However, analytic planning is itself a challenging capability. We observe that weaker models, such as Qwen-2.5-7B-Instruct (Bai et al., 2023), often fail to produce reliable plans. As shown in Fig. 1c, low-quality plans can degrade downstream CoT and final answers, whereas plans generated by stronger models (e.g., GPT-o1) lead to consistent performance gains. This contrast indicates that reasoning improvements critically depend on the quality of analytic plans rather than the mere presence of planning. However, existing plan–action based approaches (Yao et al., 2022; Wang et al., 2023; Wan et al., 2025b; Dou et al., 2026; Liang et al., 2026b) emphasize structural separation between planning and execution, without explicitly formulating plan quality as a reward signal for policy optimization.

These observations suggest that analytic planning should

be treated as a first-class optimization target. A natural approach is to leverage reinforcement learning (RL), which enables trajectory-level, non-differentiable optimization and can align plan generation with downstream reasoning behavior. However, existing outcome-based RL with verifiable rewards (RLVR) methods—such as GRPO (Shao et al., 2024) and DAPO (Yu et al., 2025)—optimize only final answer correctness, ignoring the quality of intermediate planning and reasoning (Fig. 2). Consequently, poorly structured plans and CoT trajectories may receive the same reward as well-organized ones, limiting the model's ability to learn robust global planning. Meanwhile, certain plan-based reasoning architectures (Yao et al., 2022; Zhou et al., 2023; Erdogan et al., 2025) explicitly decouple the planning and execution stages. Such a design makes it difficult to obtain reliable and verifiable reward signals for the planning component itself, as its quality cannot be directly or independently evaluated.

To address this limitation, we propose **PTA-GRPO** (**p**lan-**t**hen-**a**ction enhanced reasoning with **G**roup **R**elative **P**olicy **O**ptimization), a two-stage plan-then-reason training framework that explicitly optimizes both analytic planning and detailed reasoning. Unlike prior works (Yao et al., 2022; Wan et al., 2025b) that treat the plan merely as an additional structural component, we formulate the plan as an explicit optimization variable. In the first stage, we introduce a Planning-Structured Reasoning cold-start strategy, where a given LLM summarizes ground-truth CoT into concise high-level guidance that captures core concepts and global reasoning structure. This guidance, paired with the original CoT, forms a supervised fine-tuning (SFT) dataset that initializes explicit planning ability—an ability largely absent from base pre-trained models (Gandhi et al., 2025;

Yue et al., 2025b; Li et al., 2025). In the second stage, we develop a plan-guidance-aware RL method based on GRPO. Unlike standard GRPO, which rewards only the final response, our approach incorporates a refined reward signal that evaluates the quality of the generated high-level guidance during reasoning. This design encourages the model to produce not only correct answers but also effective and precise analytic plans, leading to more stable and globally guided reasoning trajectories. Through rigorous theoretical derivations and empirical validation, we demonstrate that the proposed reinforcement learning method can significantly increase the mutual information between predicted answers and ground-truth answers, thereby improving the accuracy of the model's generated outputs. Experimental results demonstrate that our method achieves significant performance improvements across multiple benchmarks and domains, and is applicable to both LLMs and MLLMs. Our main contributions are summarized as follows:

- **A novel two-stage plan-reasoning framework:** We propose PTA-GRPO, a two-stage training framework, including high-level guidance planning and guidance-aware reinforcement learning, to foster explicit higher-order planning and reasoning abilities in LLMs.

- **High-level guidance as supervision signal:** In the supervised fine-tuning stage, we leverage a given LLM to summarize raw chain-of-thought (CoT) into concise high-level guidance, which is combined with the original CoT, providing stronger initialization for reasoning.

- **Impressive reasoning performance:** We evaluate the proposed method across five different models, ten benchmarks, and a variety of domains and modalities. Experimental results show that it achieves significant improvements in the vast majority of settings and attains excellent performance in most evaluated scenarios.

## 2. Preliminaries and related work

### 2.1. Reasoning in Large Language Models

The reasoning of an LLM can be formalized as a token-level Markov Decision Process (MDP) (Ouyang et al., 2022; Wan et al., 2025a; Liu et al., 2025), where the state is the context sequence, the action is the next token, and the policy is the model's conditional distribution. Given a question $q$, a response $\mathfrak{o} = [\mathfrak{o}^1, \ldots, \mathfrak{o}^T]$ is sampled step by step from $\pi_\theta(\cdot \mid q, \mathfrak{o}^{<t})$. Current inference typically relies on CoT, producing a reasoning chain $c$ and final answer, but this purely autoregressive process lacks global planning, often leading to redundancy and incoherence (Wan et al., 2025a).

### 2.2. Group Relative Policy Optimization and Its Extensions

GRPO (Shao et al., 2024), proposed by DeepSeek, enhances LLM reasoning without value models by sampling multiple responses per prompt and using the group average reward as a baseline. This simple mechanism has proven effective in mathematical reasoning, code generation, and QA. Subsequent variants refine GRPO from different perspectives: SRPO (Zhang et al., 2025b) reuses samples via history resampling; DAPO (Yu et al., 2025) filters extreme cases with dynamic sampling; Dr.GRPO (Liu et al., 2025) mitigates length bias; EMPO (Zhang et al., 2025a) optimizes semantic entropy directly; and SEED-GRPO (Chen et al., 2025) integrates entropy as an uncertainty measure for more conservative updates. While these methods substantially improve mathematical reasoning, they do not explicitly target higher-order reasoning abilities. GiGPO (Feng et al., 2025) introduces a hierarchical episode-level and step-level relative advantage estimation within the group-based reinforcement learning framework, enabling fine-grained credit assignment for multi-turn LLM agents while significantly improving long-horizon decision-making performance without requiring additional rollouts or critic networks.

### 2.3. Motivation

To address the lack of global guidance in LLM reasoning, which often leads to redundancy or off-topic reasoning, inspired by human thinking habits for complex tasks or problems (Kahneman, 2011; Kahneman & Tversky, 2013), we introduce a concise high-level plan $t$ as an outline before generating the detailed CoT $c$ and its corresponding answer. Formally, the model's output can be represented as $\mathfrak{o} = t, c$, where $t$ provides the overall problem-solving direction without involving concrete computational steps, and $c$ is then generated conditioned on both the question $q$ and the plan $t$, i.e., $c = \pi_\theta(\cdot \mid q, t)$. The CoT $c$ and its final answer are guided by the high-level plan $t$. This *plan-then-reason* mechanism equips the reasoning process with global guidance, leading to more concise and accurate CoT.

Therefore, in GRPO optimization (the formulas are shown in Appendix A.7) in our study, the objective goes beyond simply ensuring the correctness of the answer in $\mathfrak{o}$. It also includes enhancing the quality of the high-level plan $t$, with the aim of producing $t$ more accurately and effectively. By improving $t$, the model receives structured guidance that can better direct the generation of the CoT $c$ and, consequently, the final answer. This dual focus rewards correct answers while strengthening high-quality intermediate reasoning, resulting in more robust and generalizable reasoning.

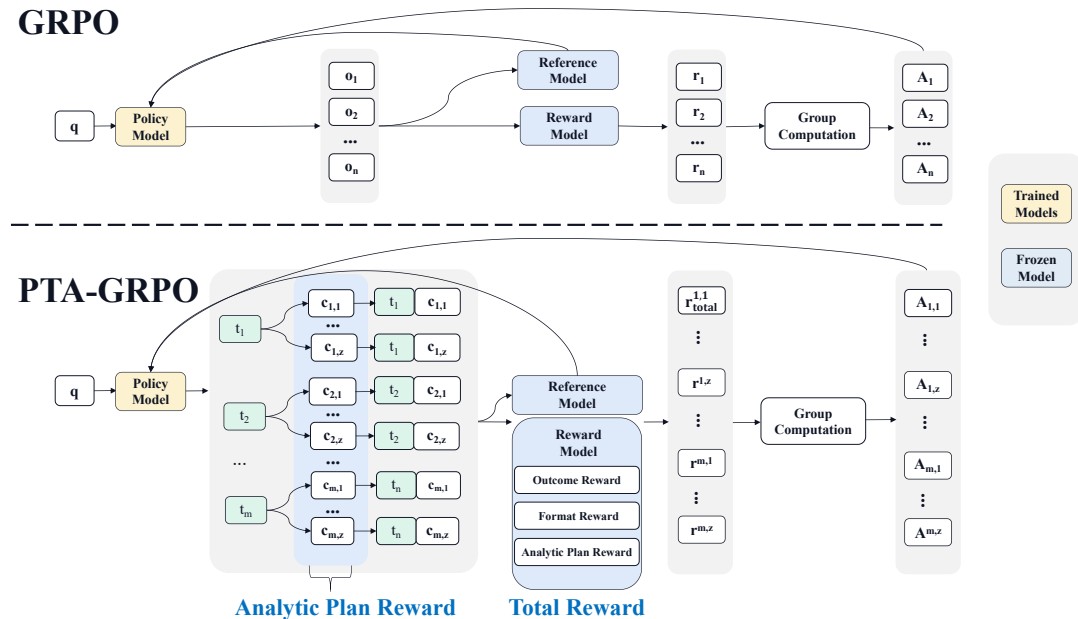

*Figure 2.* Comparison between GRPO and PTA-GRPO. It is worth noting that, to ensure a fair comparison, the number of rollout responses is kept the same between GRPO and PTA-GRPO.

## 3. Approach of PTA-GRPO

We introduce the PTA-GRPO training framework, which consists of two key components. **(1) Planning Structured Reasoning Cold-Start (PSR-CS).** This module adopts an SFT-based cold-start strategy by explicitly introducing a general analytic plan before detailed reasoning, providing high-level guidance for subsequent reasoning and answer generation. **(2) Planning Structure-Guided Reinforcement Learning (PSG-RL).** We propose a GRPO-based structure-guided reinforcement learning framework that explicitly models the quality of analytic plan as a reward signal and incorporates it into the optimization objective, thereby improving the model's structured reasoning capability and answer generation performance.

### 3.1. Planning Structured Reasoning Cold-Start (PSR-CS)

**Analytical-Guided SFT Dataset Construction.** For LLMs, the ability to perform effective planning plays a critical role in problem solving. However, existing supervised fine-tuning (SFT) datasets typically include only chain-of-thought (CoT) reasoning and final answers, while overlooking the importance of high-level analytic plan prior to reasoning. To address this limitation, we propose an analytical-guided dataset consisting of three components: the problem, a general analytic plan, and the corresponding response containing CoT reasoning and the final answer. Formally, the dataset is defined as $D_{\text{PSR-CS}} = \{(q_i, t_i, c_i)\}_{i=1}^n$, where $q_i$ denotes the problem, $t_i$ represents the general

analytic plan, and $c_i$ contains the detailed reasoning process and final answer. Unlike directly producing CoT reasoning (Wei et al., 2022), which often lacks global guidance, SFT explicitly injects the high-level problem-solving plan $t_i$ during training, enabling the model to learn the conditional distribution $c_i = \pi_\theta(\cdot \mid q_i, t_i)$ and to leverage global information when generating reasoning chains. In our dataset, the analytic plan $t_i$ is enclosed within `<plan>...</plan>` tags, while the reasoning and answer in $c_i$ are structured using `<think>...</think>` and `<answer>...</answer>` tags, respectively, forming a hierarchical representation of planning, reasoning, and answering. In contrast to prior approaches that require multi-turn interactions to obtain high-level guidance (Yao et al., 2022; Zhou et al., 2023), our design integrates planning and reasoning–answering into a single compact response, allowing the model to complete planning and execution in one pass and enabling efficient joint optimization of plan–action components under RLVR in Section 3.2.

In practice, we sample 10K instances from the Openthoughts dataset (Guha et al., 2025) as the base data. Then, a general analytic plan $t_i$ will be summarized from the given problem $q_i$ and its corresponding reasoning $c_i$ by employing an existing LLM (e.g., Qwen3-235B (Yang et al., 2025)). Notably, as the target of this process is to simply summarize the reasoning steps to a general plan, it will not heavily rely on the reasoning capability of the LLM. To verify this nature, the experiments in Table 7 (Appendix A.3) demonstrate that the plan summarized by the given LLM itself also produces the obvious improvement.

**SFT-based Cold-Start Initialization Optimization.** At this stage, we inject structured reasoning capabilities into the initial policy model $\pi_\theta$ through SFT. Specifically, we optimize the model parameters by minimizing the discrepancy between the model outputs and the reference outputs in the analytical-guided dataset $D_{\text{SRCS}}$, enabling the model to gradually learn structured reasoning patterns. The fine-tuning process is formulated as:

$$\theta_{\text{SFT}} = \min_\theta \quad \mathbb{E}_{(q_i, t_i, c_i) \in \mathcal{D}_{\text{SRCS}}} \left[ -\sum_{i=1}^{n} \log \left( \pi_\theta(t_i, c_i \mid q_i) \right) \right], \tag{1}$$

where $\theta_{\text{SFT}}$ denotes the parameter set obtained through supervised fine-tuning, and $\pi_{\theta_{\text{SFT}}}$ represents the resulting policy model equipped with structured reasoning capabilities. Through SFT-based Cold-Start Initialization, we inject planning capabilities into the policy model in a direct manner, thereby expanding its knowledge and capacity for analytic planning and providing effective supervision signals (Shah et al., 2025) for accurate answer generation.

## 3.2. Plan Structure-Guided Reinforcement Learning (PSG-RL)

After obtaining the policy model $\pi_{\theta_{\text{SFT}}}$ from the SFT stage, the RL phase then focuses on improving the model's planning capability and ensuring its effective execution. At this stage, we not only consider the correctness of CoT $c$ and its answer as part of the reward signal, but also evaluate the quality of the analytic plan $t$, which is incorporated as another important aspect of the reward signal.

### 3.2.1. ANALYTICAL PLAN–GUIDED REWARD AUGMENTATION IN GRPO

In PTA-GRPO, we design a composite reward function that integrates three aspects: the analytic plan reward ($r_{\text{analytic}}$) to encourage structured reasoning plans, the outcome accuracy reward ($r_{\text{outcome}}$) to ensure correct final results, and the structured format reward ($r_{\text{format}}$) to enforce clear and consistent output. Together, these rewards are combined into the final total reward $R_{\text{total}}$.

**Analytical Plan Reward.** Since directly evaluating the quality of an analytic plan $t$ is infeasible, we adopt a computable surrogate objective that measures how likely a plan can guide a CoT reasoning process toward the correct answer. We define the analytic plan reward $r_{\text{analytic}}$ as this probability, which serves as a proxy for plan quality.

Given a question $q$, we construct a response group $G$ via a two-stage sampling process. First, the policy model samples $m$ candidate analytic plans $\{t_i\}_{i=1}^{m}$, where $t_i \sim \pi_\theta(\cdot \mid q)$ and each $t_i$ is a concise text-based outline for solving $q$. Then, following (Lu et al., 2025), for each plan $t_i$ we resample $z$ detailed CoT trajectories $\{c_{i,k}\}_{k=1}^{z}$ under its

guidance, where $c_{i,k} \sim \pi_\theta(\cdot \mid t_i, q)$. The resulting response group is

$$G = \left\{ \{(t_i, c_{i,k})\}_{k=1}^{z} \right\}_{i=1}^{m}.$$

Each element in $G$ forms a plan–CoT pair. The reward assigned to an analytic plan $t_i$ is defined as the empirical accuracy of its resampled outcomes:

$$s_i = \frac{1}{z} \sum_{k=1}^{z} \mathbb{I}[\hat{y}_{i,k} = y], \quad r_{\text{analytic}}(t_i) = [\text{softmax}(\mathbf{s}/\tau)]_i, \tag{2}$$

where $\mathbb{I}[\cdot]$ is the indicator function, $\hat{y}_{i,k}$ denotes the final predicted answer extracted from $c_{i,k}$, and $y$ is the ground-truth answer for $q$. The Softmax operation amplifies relative score differences, highlighting high-quality plans while suppressing low-quality ones. Importantly, the CoT set $\{c_{i,k}\}_{k=1}^{z}$, used to assess analytic plan $t_i$ quality, is also incorporated into the policy update in Group $G$, enabling a dual-use mechanism without extra sampling overhead. Compared to RLVR methods with the same rollout budget (e.g., GRPO), our approach does not incur noticeable additional time cost. RL training time is detailed in Appendix A.2.

Optimizing the policy with $r_{\text{analytic}}(\cdot)$ encourages the generation of more accurate analytic plans and increases the probability of correct prediction $\Pr(\hat{y} = y \mid t, q)$. In contrast to traditional RLVR methods (Yu et al., 2025; Feng et al., 2025), which rely solely on outcome-based supervision and cannot directly supervise intermediate reasoning, the analytic plan reward $r_{\text{analytic}}$ enables direct assessment of intermediate reasoning trajectories and assigns higher rewards to those more likely to succeed. Section 3.3 shows, both theoretically and empirically, that optimizing $r_{\text{analytic}}$ increases the mutual information between $y$ and $\hat{y}$, thereby enhancing reasoning ability.

**Outcome Reward.** The outcome reward, defined as $r_{\text{outcome}}$, is a result-based terminal reward similar to GRPO, used to evaluate whether the predicted answer aligns with the ground truth. For each plan–CoT response $(t_i, c_{i,k})$, the outcome reward $r_{\text{outcome}}$ is defined as follows:

$$r_{\text{outcome}} = \begin{cases} 1, & \hat{y}_{i,k} = y, \\ 0, & \text{else}. \end{cases} \tag{3}$$

The outcome reward $r_{\text{outcome}}$ encourages the policy model to learn to follow the analytic plan $t_i$ and to develop the ability to generate correct answers that strive for correctness.

**Format Reward.** To ensure structural consistency and concise outputs, we incorporate a format reward $R_{\text{format}}$ that enforces a predefined response template and discourages overly long responses. The detailed formulation is in the Appendix A.1.

**Total Reward.** The above three rewards together constitute the total reward $R_{\text{total}}$ for each response as:

$$R_{\text{total}} = R_{\text{outcome}} + \beta \cdot R_{\text{analytic}} + R_{\text{format}}, \quad (4)$$

where $\beta$ represents the hyperparameter. We first obtain a total reward set $\{\{r_{total}^{i,k}\}_{i=1}^m\}_{k=1}^z$, where $r_{total}^{i,k}$ denotes the total reward of the $k$-th CoT generated under the guidance of the $i$-th analytic. Based on this reward, we compute the corresponding advantage function $A_{i,k}$ using Eq. 10, and subsequently incorporate it into the update rule in Eq. 9 to optimize the model.

*Table 1.* Performance comparison of different post-training methods using various base models. **Bold** is best per block.

| Method | MATH500 | AIME24 | AIME25 | AMC23 | Average |
|---|---|---|---|---|---|
| **Qwen2.5-7B-Instruct** | 63.93 | 14.25 | 3.77 | 54.95 | 34.23 |
| GRPO | 90.67 | 30.15 | 25.20 | 73.21 | 54.81 |
| DAPO | 91.87 | 31.64 | 22.53 | 74.12 | 55.04 |
| CPL | 87.53 | 28.55 | 24.49 | 72.53 | 53.28 |
| Full-Step-DPO | 89.25 | 27.85 | 23.39 | 69.52 | 52.50 |
| ORZ | 88.55 | 30.44 | 26.27 | 72.93 | 54.55 |
| PTA-GRPO | **92.53** | **34.53** | **29.25** | **77.51** | **58.46** |
| **LLaMA3.2-3B** | 36.64 | 3.97 | 2.88 | 19.68 | 15.79 |
| GRPO | 62.23 | 19.55 | 18.29 | 46.44 | 36.63 |
| DAPO | 63.97 | 18.97 | 18.55 | 46.44 | 36.98 |
| PTA-GRPO | **66.25** | **24.92** | **20.75** | **49.55** | **40.37** |
| **Qwen3-8B** | 90.65 | 65.27 | 52.33 | 88.58 | 74.21 |
| GRPO | **94.33** | 71.57 | 54.94 | 93.51 | 78.59 |
| DAPO | 93.09 | 69.17 | 52.77 | 92.35 | 76.85 |
| CPL | 92.22 | 70.79 | 52.33 | 91.52 | 76.72 |
| Full-Step-DPO | 93.03 | 71.78 | 50.65 | 92.37 | 76.96 |
| ORZ | 93.95 | 68.67 | 53.97 | 92.27 | 77.22 |
| PTA-GRPO | 93.79 | **72.68** | **56.14** | **93.77** | **79.10** |
| **Qwen3-14B** | 91.75 | 72.39 | 70.55 | 93.97 | 82.17 |
| GRPO | 92.48 | 73.62 | 71.33 | 94.87 | 83.08 |
| DAPO | 93.27 | 72.55 | 71.77 | **95.66** | 83.31 |
| PTA-GRPO | **96.15** | **75.44** | **73.29** | 95.17 | **85.01** |

**Advantages of PTA-GRPO.** Compared with standard GRPO, which primarily relies on sparse task-level accuracy supervision, our guidance-aware PTA-GRPO framework introduces several critical improvements. **First**, powered by the analytic-plan reward $r_{\text{analytic}}$, the model gains the ability to *evaluate* its intermediate reasoning process, which RLVR cannot achieve with purely outcome-based signals. This mechanism drives the model to construct higher-level analytic plans and use them to guide more reliable CoT reasoning. **Second**, the outcome reward $r_{\text{outcome}}$ encourages the policy model to follow the analytic plan and enhance its reasoning capability under such structured guidance. Besides, following (Gandhi et al., 2025), PTA-GRPO enhances LLM self-reflection in reinforcement learning by adjusting the prompt in Appendix A.11, allowing the model to correct later steps even when the initial plan is flawed.

### 3.3. Theoretical Performance Analysis

In this section, we theoretically analyze the impact of optimizing $r_{\text{analytic}}$ on the error probability of the policy model.

Our theoretical findings are as follows.

**Theorem 3.1.** *Let $q$ denote the input question, $t$ the analytic plan, $\hat{y}$ the answer predicted by the policy model, and $y$ the ground-truth answer. With error probability $p_{error}$, it holds that:*

$$p_{error} \leq \frac{1}{2}\big[H(y) - I(\hat{y}, y \mid t, q)\big], \quad p_{error} = \Pr(y \neq \hat{y}),$$

*where $H(\cdot)$ denotes the entropy, $I(\cdot)$ denotes the mutual information.*

The proof can be seen in appendix A.8. Leveraging the conclusion from (Qian et al., 2025), since $H(y)$ depends solely on the fixed distribution of the answer and is independent of the model's reasoning steps, it can therefore be regarded as a constant. In our Theorem 3.1, the upper bound of the error probability $p_{\text{error}}$ is governed by the conditional mutual information $I(\hat{y}; y \mid t, q)$, which measures the statistical dependence between the predicted output $\hat{y}$ and the true label $y$, given the auxiliary analytic plan $t$. In other words, the larger the shared information between $\hat{y}$ and $y$ under the guidance of the analytic plan $t$, the tighter the upper achievable limit on the probability of error.

**Proposition 3.2.** *Let $t_1$ and $t_2$ be any analytic plans, and let $r_{\text{analytic}}(t_1)$ and $r_{\text{analytic}}(t_2)$ denote their corresponding analytic rewards. Let $\hat{y}_1$ and $\hat{y}_2$ be the answers induced by executing $t_1$ and $t_2$, respectively. If $r_{\text{analytic}}(t_1) \geq r_{\text{analytic}}(t_2)$, it holds that*

$$H(y|\hat{y}_1, t_1, q) \leq H(y|\hat{y}_2, t_2, q). \quad (5)$$

*Remark* 3.3. By the definition of mutual information, $I(\hat{y}; y \mid q, t) = H(y \mid q, t) - H(y \mid \hat{y}, q, t)$. Note that $H(y \mid q, t)$ is solely determined by the underlying data distribution of $(q, t, y)$ and is independent of the model's prediction $\hat{y}$. Hence, $H(y \mid q, t)$ can be regarded as a constant with respect to the learning or inference process. As shown in Proposition 3.2, as the analytic plan reward $r_{\text{analytic}}$ increases, the conditional entropy $H(y \mid \hat{y}, q, t)$ decreases, which in turn implies a larger mutual information $I(y; \hat{y} \mid q, t)$ between the prediction $\hat{y}$ and the ground-truth $y$. In particular, we have $\max_t r_{\text{analytic}}(t) \iff \max_t I(y; \hat{y} \mid q, t)$. Therefore, optimizing the analytic plan reward $r_{\text{analytic}}$ effectively improves the model's reasoning ability.

**Empirical Analysis.** To assess the reliability of the above theory, we further examine the relationship between mutual information and the plan reward. Following (Qian et al., 2025), we use the Hilbert–Schmidt Independence Criterion (HSIC) to estimate the mutual information between the predicted answer $\hat{y}$ and the ground-truth answer $y$. As shown in Fig. 3, the plan reward and the mutual information exhibit similar increasing trends, which is consistent with our theoretical claim $\max_t r_{\text{analytic}}(t) \iff \max_t I(y; \hat{y} \mid q, t)$ and thus supports its validity.

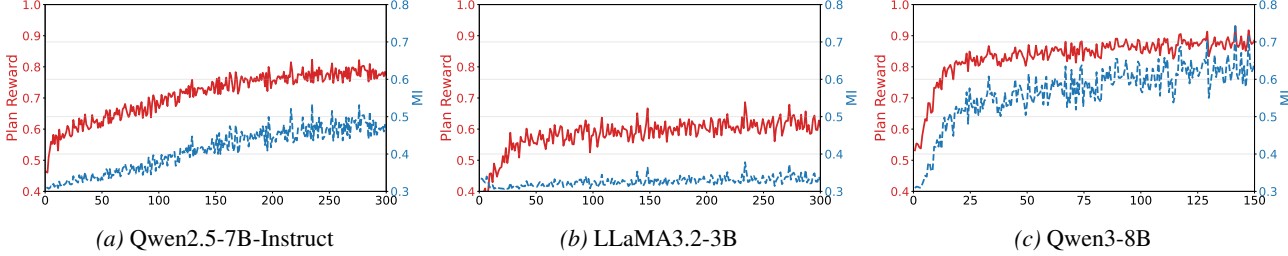

*(a) Qwen2.5-7B-Instruct*       *(b) LLaMA3.2-3B*       *(c) Qwen3-8B*

*Figure 3.* Trend plots of mutual information and plan reward across different models.

*Table 2.* Impact of data scale of RL on PTA-GRPO, where Qwen2.5-7B-Instruct is considered as base model. **Bold** is best per block.

| Data scale | MATH500 | AIME24 | AIME25 | AMC23 | Average |
|---|---|---|---|---|---|
| 4k | 82.27 | 27.22 | 21.03 | 65.22 | 48.94 |
| 8k | 83.59 | 28.23 | 22.29 | 68.29 | 50.60 |
| 11k | 84.23 | 29.33 | 24.51 | 69.37 | 51.86 |
| 14k | 85.57 | 30.26 | 25.97 | 70.24 | 53.01 |
| 30k | 88.45 | 31.45 | 26.29 | 75.29 | 55.37 |
| 60K | **94.53** | **34.53** | **29.25** | **77.51** | **58.46** |

## 4. Experiment

**Base Models.** To evaluate PTA-GRPO, we adopt four base models of varying scales and series: LLaMA3.2-3B (Dubey et al., 2024), Qwen2.5-7B-Instruct (Bai et al., 2025), Qwen3-8B, and Qwen3-14B (Yang et al., 2025), enabling a comprehensive assessment of its robustness across architectures. Training details are in Section A.10. For our vision-language model (VLM), we use Qwen2.5-7B-VL (Bai et al., 2025) as the base model to extend our experiments.

**Training Datasets and Benchmarks.** For SFT, we use 10K samples from Openthoughts (Guha et al., 2025) with injected planning knowledge (Section 3.1). For RL, we sample 60K problems from DeeMath (He et al., 2025), which offers graded difficulty and is rigorously decontaminated to avoid benchmark leakage. We evaluate PTA-GRPO on AIME24, AIME25, MATH500, and AMC23, and report the average accuracy over 64 independent runs. Besides, we assess it on the general-purpose multimodal datasets MMMU-Pro (Yue et al., 2025a), MMMU (Yue et al., 2024), and EMMA (Standley et al., 2023), and the scientific benchmark dataset MMK-12 (Meng et al., 2025).

*Table 3.* Ablation analysis on PTA-GRPO, where Qwen2.5-7B-Instruct is considered as base model. **Bold** is best per block.

| Method | MATH500 | AIME24 | AIME25 | AMC23 | Average |
|---|---|---|---|---|---|
| PTA-GRPO $_{w/o\ SFT}$ | 84.27 | 20.53 | 18.39 | 62.13 | 46.33 |
| PTA-GRPO $_{w/o\ r_{format}}$ | 91.19 | 33.75 | 28.74 | 75.58 | 57.32 |
| PTA-GRPO $_{w/o\ r_{analytic}}$ | 88.76 | 29.48 | 25.53 | 74.47 | 54.56 |
| PTA-GRPO | **92.53** | **34.53** | **29.25** | **77.51** | **58.46** |

**Baseline.** We compare PTA-GRPO with the base model

and several RLVR like GRPO (Shao et al., 2024), and DAPO (Yu et al., 2025). In addition, we compare our method with several advanced reinforcement learning algorithms, including CPL (Wang et al., 2024b), Full-Step DPO (Xu et al., 2025b), and ORZ (Hu et al., 2025). For fairness, all methods use the same SFT and RL data (differing only in the improved SFT portion). For a fair comparison, we use the same number of sampled responses as all selected RLVR methods, so their time consumption is nearly the same.

### 4.1. Performance of PTA-GRPO

Table 1 shows that our method (PTA-GRPO) consistently outperforms both the base models and other RLVR approaches across different model scales and evaluation benchmarks. For relatively weaker backbones such as Qwen2.5-7B-Instruct and LLaMA3.2-3B, PTA-GRPO delivers the most significant improvements, raising the average scores by over 20 points compared to the raw models and further surpassing GRPO and DAPO by clear margins. PTA-GRPO demonstrates strong effectiveness. Detailed analysis is provided in Appendix A.6.

*Table 4.* The impact of datasets containing analytic planning on SFT. **Bold** is best per block.

| Base Model | Method | MATH500 | AIME24 | AIME25 | AMC23 | Average |
|---|---|---|---|---|---|---|
| Qwen2.5-7B-Instruct | SFT w/o planning | 78.28 | 21.66 | 19.66 | 60.53 | 45.03 |
| | SFT w/ planning | **80.40** | **25.25** | **20.33** | **63.75** | **47.43** |
| Qwen3-8B | SFT w/o planning | 91.02 | 70.03 | 50.25 | 92.39 | 75.92 |
| | SFT w/ planning | **92.53** | **71.97** | **51.77** | **93.55** | **77.46** |

### 4.2. Impact of RL Data Scaling

Table 2 shows that, with Qwen2.5-7B-Instruct as the base model, scaling the RL training data from 4k to 60k leads to consistent improvements for PTA-GRPO across four math benchmarks (MATH500 / AIME24 / AIME25 / AMC23), indicating that more RL data yields stable performance gains. Notably, the improvement is not strictly linear: the gains are stronger at some stages and smaller at others, while a larger-scale setting still brings a clear boost. This suggests that scaling up the RL data continues to unlock additional capability rather than quickly saturating.

## 4.3. Ablation Analysis

Table 3 presents an ablation study of PTA-GRPO on Qwen2.5-7B-Instruct. The full PTA-GRPO achieves the best performance across all benchmarks (58.46 average), while removing SFT causes a substantial drop (46.33). Disabling either $r_{\text{format}}$ or $r_{\text{analytic}}$ also consistently hurts performance, with a larger degradation observed when removing $r_{\text{analytic}}$, indicating that both rewards contribute to final gains.

*Table 5.* Comparison between PTA-GRPO and other approaches on General-Benchmark and Science Benchmark, using Qwen2.5-7B-VL as the base model.

| Method | MMMU-Pro | MMMU | EMMA | Phys | Chem | Bio |
|---|---|---|---|---|---|---|
| **Base** | 36.9 | 54.3 | 21.5 | 45.4 | 56.4 | 54.0 |
| SRPO | 42.3 | 57.1 | 29.6 | 56.2 | 65.2 | 65.2 |
| PTA-GRPO | **44.7** | **59.0** | **31.9** | **58.5** | **68.7** | **66.8** |

## 4.4. Impact of Analytic Plan on SFT

Table 4 compares standard SFT (w/o planning) with SFT on $\mathcal{D}_{\text{SRCS}}$ augmented by analytic plans (w/ planning). Across both base models, SFT with analytic planning consistently outperforms SFT without planning on all four benchmarks: for Qwen2.5-7B-Instruct, the average score rises from 45.03 to 47.43 (+2.40), and for Qwen3-8B, from 75.92 to 77.46 (+1.54), with gains appearing simultaneously on MATH500, AIME24/25, and AMC23 rather than as isolated fluctuations. This suggests that plans provide an important supervision signal, teaching the model transferable solution organization and reasoning trajectories that yield robust improvements across tasks and models.

## 4.5. Empirical Evaluation on Generalization

Beyond mathematics, we also evaluate on multimodal (Table 5), general-domain, and scientific benchmarks, including MMMU-Pro (Yue et al., 2025a), MMMU (Yue et al., 2024), EMMA (Standley et al., 2023), and MMK-12 (Meng et al., 2025). Using SRPO (Wan et al., 2025a) as baselines and following SRPO's SFT/RL data for cold-start and training, PTA-GRPO with Qwen2.5-7B-VL consistently outperforms the Base model and SRPO on MMMU-Pro, MMMU, EMMA, and Phys/Chem/Bio benchmarks, achieving uniformly better metrics and stronger generalization reasoning.

## 4.6. Results of test-time scaling

We next examine the effectiveness of PTA-GRPO under multiple sampling at test time. As shown in Fig. 4, PTA-GRPO consistently outperforms GRPO on the AIME2025 dataset across Pass@1, Pass@4, Pass@8, and Pass@16. This demonstrates that PTA-GRPO maintains high precision under low-sample conditions, while further exhibiting stronger solution coverage as the number of sam-

ples increases.

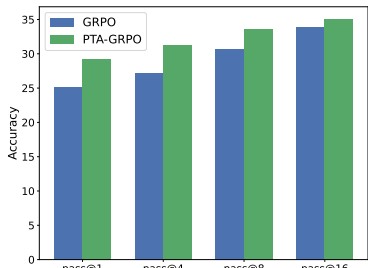

*Figure 4.* Effect of scaling test-time compute on AIME25 (Pass@K), with Qwen2.5-7B-Instruct as the base model.

## 4.7. Sensitivity analysis

From Table 6, it can be observed that the parameter $\beta$ has a notable impact on the overall performance of the algorithm PTA-GRPO, while the sensitivity varies across different datasets. Overall, the best Average performance (58.46) is achieved when $\beta = 0.5$, indicating that this setting provides a favorable balance across multiple tasks.

Specifically, performance on MATH500 shows a trend of first increasing and then decreasing as $\beta$ grows, reaching its peak at $\beta = 0.7$ (93.07), which suggests that this task benefits from a relatively larger $\beta$. In contrast, AIME24 and AIME25 achieve their best results at $\beta = 0.5$, and their performance drops noticeably when $\beta$ is either increased or decreased, indicating higher sensitivity to the parameter choice. For AMC23, the highest score is obtained at a smaller value, $\beta = 0.3$ (77.98), after which the performance gradually declines as $\beta$ increases, reflecting a different preference from the other datasets.

*Table 6.* Sensitivity analysis of PTA-GRPO

| $\beta$ | MATH500 | AIME24 | AIME25 | AMC23 | Average |
|---|---|---|---|---|---|
| 0.3 | 91.95 | 33.28 | 28.57 | **77.98** | 57.95 |
| 0.5 | 92.53 | **34.53** | **29.25** | 77.51 | **58.46** |
| 0.7 | **93.07** | 30.51 | 26.29 | 75.45 | 56.33 |
| 0.9 | 89.04 | 28.82 | 27.53 | 74.48 | 54.97 |

In summary, the choice of $\beta$ involves a trade-off among different tasks: larger $\beta$ values favor MATH500, while a moderate setting ($\beta = 0.5$) yields the most robust overall performance. Therefore, when considering multiple datasets simultaneously, $\beta = 0.5$ represents a reasonable choice.

To provide a more comprehensive evaluation, we conduct a detailed sensitivity analysis of the number of plans $m$ and the number of CoT trajectories $z$ sampled per plan. The full results and discussions are presented in Appendix A.5. The sensitivity results for parameter $m$ are shown in Fig. 7, whereas the analysis for parameter $z$ is summarized in Table 8.

# 5. Conclusion

We propose Plan-Guide Enhanced Reasoning with Group Relative Policy Optimization (PTA-GRPO), which integrates high-level planning with fine-grained reasoning to alleviate the lack of global planning in traditional CoT reasoning. Experimental results show that PTA-GRPO achieves significant improvements across multiple mathematical reasoning benchmarks and model scales, validating its effectiveness and generalization.

# Impact Statement

This paper introduces PTA-GRPO, a two-stage training framework that explicitly optimizes high-level analytic planning in large language model (LLM) reasoning. By incorporating plan quality as a reinforcement learning objective alongside final-answer correctness, the proposed method aims to improve global reasoning coherence, reduce redundant or inconsistent reasoning trajectories, and enhance generalization across tasks and modalities.

From a positive perspective, improving structured planning and reasoning may contribute to more reliable LLM behavior in domains that require multi-step problem solving, such as mathematics, science education, programming, and multimodal reasoning. By encouraging models to produce explicit intermediate plans and better-aligned reasoning trajectories, our method may also improve interpretability compared to purely outcome-based reinforcement learning approaches.

However, stronger reasoning and planning capabilities can also amplify potential risks associated with advanced LLM systems. More effective structured reasoning may enable models to solve increasingly complex tasks, which could be misused in contexts involving academic dishonesty, automated generation of misleading content, or assistance in harmful problem-solving scenarios. Although our work focuses on benchmarked reasoning tasks in mathematics and science, the underlying techniques are general and could be applied to broader domains.

Importantly, PTA-GRPO does not introduce new data sources, external tools, or autonomous capabilities beyond standard language model training. It modifies the training objective to better align intermediate planning with correct outcomes. The method remains compatible with existing alignment, safety filtering, and content moderation techniques. We believe that responsible deployment, continued alignment research, and domain-specific safeguards are necessary to mitigate misuse risks as reasoning capabilities improve.

Overall, this work aims to advance research on structured reasoning in LLMs while acknowledging that improvements in reasoning power should be accompanied by careful consideration of downstream societal impacts and appropriate governance mechanisms.

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

# A. Appendix

## A.1. Details of Format Reward

**Format Reward.** The format reward $r_{\text{format}}$ is designed to regulate the overall structure of the model response, ensuring both conformity to the desired format and control over the output length. It consists of two components: $r_{\text{structure}}$ and $r_{\text{length}}$. Specifically, $r_{\text{structure}}$ enforces that the policy model's response adheres to the predefined structural template, i.e., `<plan>...</plan>`, `<think>...</think>`, and `<answer>...</answer>`. Meanwhile, $r_{\text{length}}$ serves as an auxiliary reward that encourages the model to generate concise and efficient token sequences, thereby reducing redundant or uninformative content.

To provide a clearer illustration of each reward, we present its detailed formulation as follows. We begin with the format reward $r_{\text{format}}$, which is defined as:

$$r_{\text{format}} = \begin{cases} 0.2, & \text{if the response strictly} \\ & \quad \text{follows the predefined template} \\ 0, & \text{otherwise.} \end{cases} \tag{6}$$

This function enforces a binary constraint on the output structure: a full reward is granted only when the response strictly adheres to the predefined template, thereby ensuring the consistency and parsability of the generated results.

For response length, the optimal number of tokens varies across different questions, making it difficult to predefine a fixed target length. Therefore, for all responses generated for a given question, we select the shortest correct response length as the reference length $T$, defined as:

$$T = \min\{\,|\{t_i, c_{i,k}\}|\ |\ \hat{y}_{i,k} = y\,\}, \tag{7}$$

where $|\{t_i, c_{i,k}\}|$ denotes the token length of response $\{t_i, c_{i,k}\}$. Here, $T$ represents the shortest executable token length required to obtain the correct answer to a given question. It can be regarded as the optimal reference length under current knowledge, toward which other correct responses should converge in order to minimize redundancy while preserving correctness. For each response $\{t_i, c_{i,k}\} \in G$, the length reward $r_{\text{length}}$ can be expressed as:

$$r_{\text{length}}(\{t_i, c_{i,k}\}) = \alpha \cdot \exp(-\frac{|\,|\{t_i, c_{i,k}\}| - T|}{T_{\max} - T}), \tag{8}$$

where $\alpha$ is a hyperparameter, and $T_{\max}$ does not denote the maximum output length set for the policy model. The reward becomes larger as the response length approaches the reference length $T$, encouraging the model to generate concise yet correct responses.

The format reward $r_{\text{format}}$, defined as $r_{\text{format}} = r_{\text{structure}} + r_{\text{length}}$, ensures that the output not only adheres to the required format, but also guarantees the conciseness of the output response.

## A.2. RL training time for various RLVR approaches

Using Qwen2.5-7B-Instruct as the base model, the training time of GRPO is 44.7 hours, DAPO requires 47.5 hours, and PTA-GRPO takes 44.9 hours. When using Qwen3-8B as the base model, GRPO requires 59.7 hours, DAPO 66.9 hours, and PTA-GRPO 61.4 hours. These results indicate that PTA-GRPO does not introduce a noticeable increase in training time compared to existing RLVR-based methods.

## A.3. Impact Results of Self-Generated Plans in a Policy Model

*Table 7.* Comparative Analysis of Self-Generated Plans in a Policy Model

| Method | MATH500 | AIME24 | AIME25 | AMC23 | Average |
|---|---|---|---|---|---|
| PTA-GRPO w/ self-plan | 92.18 | 32.79 | 28.57 | 75.94 | 57.37 |
| PTA-GRPO | 92.53 | 34.53 | 29.25 | 77.51 | 58.46 |

Table 7 compares the performance of PTA-GRPO with and without self-generated plans, where 'PTA-GRPO w/ self-plan' denotes that the plan is summarized and generated solely by the policy model without relying on an advanced LLM

for planning. The results show that the two settings exhibit highly consistent performance across all benchmarks: the difference on MATH500 is negligible, while slight performance gaps appear on more challenging tasks such as AIME24, AIME25, and AMC23, though the margins remain limited. In terms of average performance, 'PTA-GRPO w/ self-plan' achieves a score of 57.37, which is close to the full version's 58.46. These findings indicate that even without explicit plan generation from an advanced LLM, PTA-GRPO can maintain stable and competitive performance by leveraging the policy model's self-generated planning capability, demonstrating the intrinsic effectiveness and practical robustness of the proposed approach.

### A.4. Training Dynamics of PTA-GRPO

Fig. 5 and Fig. 6 illustrate the training dynamics of QWEN3-8B and QWEN2.5-7B-Instruct, respectively. As shown in the figures, our method outperforms GRPO in terms of accuracy reward and response length, indicating the effectiveness of the introduced component. It is worth noting that in Fig. 5 (b), our approach achieves lower entropy compared to GRPO. This suggests that for stronger models, our method encourages the development of more reasonable analytic plans, enabling the model to complete a given trajectory with greater confidence and ultimately achieving higher accuracy.

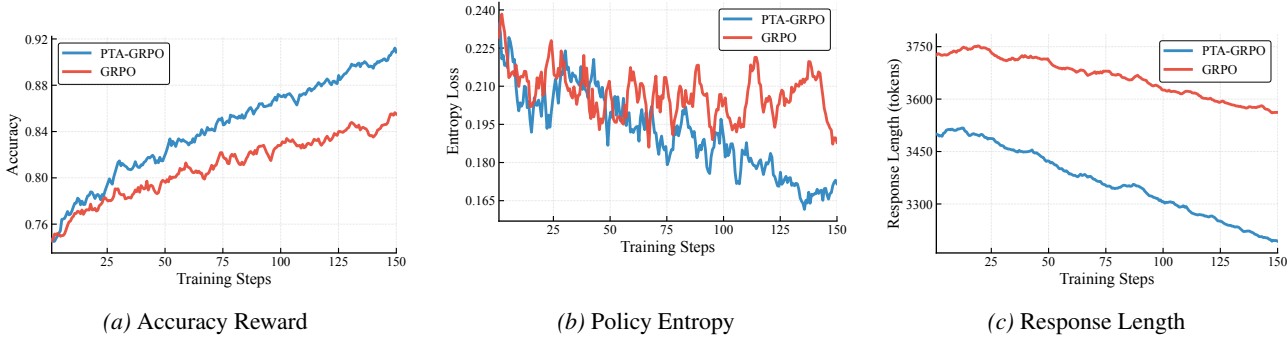

*(a)* Accuracy Reward          *(b)* Policy Entropy          *(c)* Response Length

*Figure 5.* Training Dynamics of PTA-GRPO with Qwen3-8B.

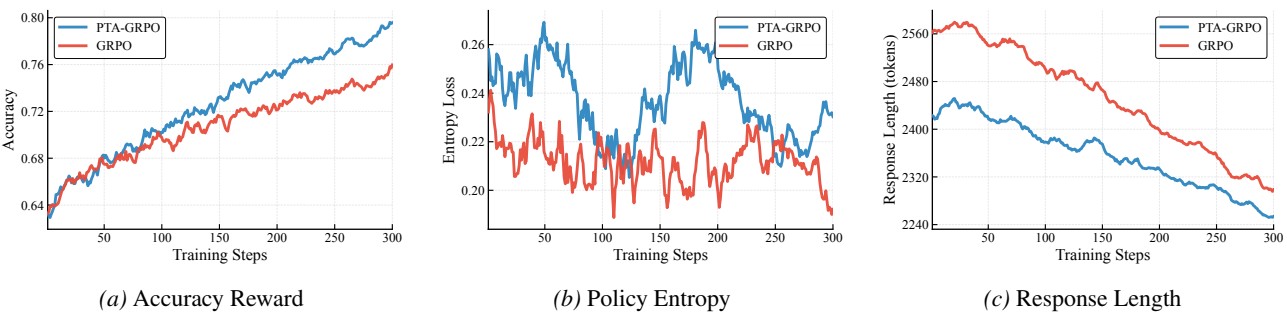

*(a)* Accuracy Reward          *(b)* Policy Entropy          *(c)* Response Length

*Figure 6.* Training Dynamics of PTA-GRPO with Qwen2.5-7B-Instruct.

### A.5. Sensitivity analysis of sampled number

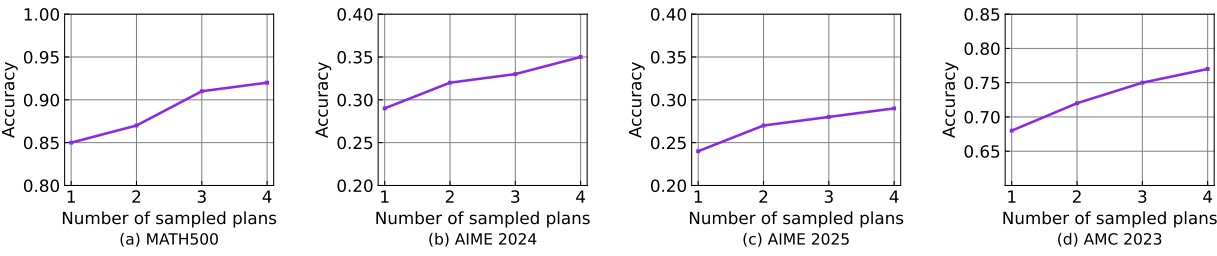

*Figure 7.* Sensitivity analysis with respect to the number of analytic plans $m$, where Qwen2.5-7B-Instruct is considered as based model.

*Table 8.* Sensitivity analysis with respect to the number of sampled CoTs $z$ for per analytic plan, using Qwen2.5-7B-Instruct as the base model.

| $z$ | MATH500 | AIME 2024 | AIME 2025 | AMC 23 | Average |
|---|---|---|---|---|---|
| 1 | 85.79 | 29.21 | 25.95 | 69.97 | 52.73 |
| 2 | 89.29 | 33.29 | 27.77 | 74.73 | 56.27 |
| 3 | 92.53 | 34.53 | 29.25 | 77.51 | 58.46 |

### A.6. Analysis of Statistical Significance

Table 9 reports pairwise significance tests comparing PTA-GRPO with the GRPO baseline across multiple models and mathematical reasoning benchmarks. Overall, PTA-GRPO yields consistent and statistically significant improvements, with 13 out of 16 model–task pairs achieving significance at the 5% level or better. The gains are particularly pronounced for smaller models, such as Qwen2.5-7B and LLaMA3.2-3B, where PTA-GRPO improves performance across all tasks with large effect sizes and strong statistical significance (all $p < 0.001$). For larger models (Qwen3-8B and Qwen3-14B), the improvements remain positive on most benchmarks but become more modest, and in some cases statistically insignificant, indicating diminishing returns as model capacity increases. Notably, the benefits of PTA-GRPO are most consistent on harder benchmarks (AIME24 and AIME25), where all evaluated models exhibit significant gains, suggesting that PTA-GRPO is particularly effective for complex, long-horizon mathematical reasoning. In contrast, on relatively easier tasks such as AMC23 and MATH500, the improvements are smaller and occasionally negligible, reflecting a saturation effect rather than a systematic degradation.

*Table 9.* Pairwise significance tests between PTA-GRPO and GRPO. Each row reports the improvement $\Delta$ (in percentage points), the $p$-value, and the corresponding significance level. *, **, and *** denote $p < 0.05$, $p < 0.01$, and $p < 0.001$, respectively; "ns" indicates not significant. ($p$-values are re-estimated using a normal-approximation scaling with $\sqrt{n}$ and $\Delta$.)

| Model | Baseline | Task | $\Delta$ (pt) | $p$ / sig |
|---|---|---|---|---|
| Qwen2.5-7B | GRPO | MATH500 | +1.86 | $0.0000^{***}$ |
| | GRPO | AIME24 | +4.38 | $0.0000^{***}$ |
| | GRPO | AIME25 | +4.05 | $0.0000^{***}$ |
| | GRPO | AMC23 | +4.30 | $0.0000^{***}$ |
| LLaMA3.2-3B | GRPO | MATH500 | +4.02 | $0.0000^{***}$ |
| | GRPO | AIME24 | +5.37 | $0.0000^{***}$ |
| | GRPO | AIME25 | +2.46 | $0.0000^{***}$ |
| | GRPO | AMC23 | +3.11 | $0.0000^{***}$ |
| Qwen3-8B | GRPO | MATH500 | -0.54 | $0.0275^{*}$ |
| | GRPO | AIME24 | +1.11 | $0.0012^{**}$ |
| | GRPO | AIME25 | +1.20 | $0.0003^{***}$ |
| | GRPO | AMC23 | +0.26 | $0.4090^{ns}$ |
| Qwen3-14B | GRPO | MATH500 | +3.67 | $0.0000^{***}$ |
| | GRPO | AIME24 | +1.82 | $0.0000^{***}$ |
| | GRPO | AIME25 | +1.96 | $0.0000^{***}$ |
| | GRPO | AMC23 | +0.30 | $0.0741^{ns}$ |

### A.7. Group Relative Policy Optimization (GRPO)

Group Relative Policy Optimization (GRPO) is a state-of-the-art Reinforcement Learning with Verifiable Rewards (RLVR) algorithm that simplifies Proximal Policy Optimization (PPO) (Schulman et al., 2017) by removing the need for a value model to estimate the baseline advantage, and has demonstrated remarkable success in enhancing the reasoning abilities of LLM. Formally, let $Q$ denote the set of questions, $\pi_{\theta_{\text{old}}}$ be the current policy model, and $\{\mathfrak{o}_i\}_{i=1}^{N}$ represent a collection of $N$ candidate responses sampled for a question $q \in Q$. We also define $\pi_{\theta_{\text{ref}}}$ as a fixed reference model. The training objective of GRPO is expressed as:

$$
\begin{aligned}
J_{\text{GRPO}}(\theta) = \mathbb{E}_{q \sim Q, \, \{\mathfrak{o}_i\}_{i=1}^{N} \sim \pi_{\theta_{\text{old}}}} \\
\left[ \frac{1}{N} \sum_{i=1}^{N} \sum_{t=1}^{|\mathfrak{o}_i|} \min\left( \frac{\pi_\theta(\mathfrak{o}_i^t|q)}{\pi_{\theta_{\text{old}}}(\mathfrak{o}_i^t|q)} A_i, \, \text{clip}\left( \frac{\pi_\theta(\mathfrak{o}_i^t|q)}{\pi_{\theta_{\text{old}}}(\mathfrak{o}_i^t|q)}, 1-\epsilon, 1+\epsilon \right) A_i \right) \right. \\
\left. - \beta D_{\text{KL}}(\pi_\theta \| \pi_{\text{ref}}) \right]
\end{aligned}
\tag{9}
$$

where $\epsilon$ controls the clipping range and $\beta$ weights the KL regularization term. The normalized advantage $A_i$ assigned to each response $o_i$ is computed from group-based rewards:

$$A_i = \frac{r_i - \mu}{\sigma}, \quad \text{with } \mu = \frac{1}{N} \sum_{j=1}^{N} r_j, \quad \sigma = \sqrt{\frac{1}{N} \sum_{j=1}^{N} (r_j - \mu)^2}, \tag{10}$$

where $\{r_1, r_2, \ldots, r_N\}$ are the scalar rewards associated with the response group $\{o_i\}_{i=1}^{N}$.

In GRPO, each response $o \in \{o_i\}_{i=1}^{N}$ consists of a CoT $c$ together with its final answer. As noted in Section 2.1, token-level MDPs lack global planning and often yield redundant steps, while GRPO rewards $r$ corresponding to $o$ focus only on final answer correctness, overlooking reasoning quality and enabling reward hacking through superficial or verbose CoTs.

### A.8. Theoretical proof

*Proof.* We consider the binary setting, where the ground-truth answer $y \in \{0, 1\}$ and $\hat{y}$ denotes a (possibly stochastic) predictor based on $(t, q)$. We set 0 is wrong and 1 is correct. All logarithms are taken base 2.

Fix $(t, q)$. Define the conditional (Bayes) error probability

$$p_c(t, q) := \Pr(\tilde{y} \neq y \mid t, q),$$

where $\tilde{y}$ denotes the Bayes-optimal (MAP) decision rule given $(\hat{y}, t, q)$, i.e.,

$$\tilde{y} := \arg \max_{y' \in \{0,1\}} \Pr(y = y' \mid \hat{y}, t, q).$$

For convenience, define the conditional posterior

$$\eta(\hat{y}; t, q) := \Pr(y = 1 \mid \hat{y}, t, q).$$

Since $y$ is binary, the conditional Bayes error given $(\hat{y}, t, q)$ is

$$\Pr(\tilde{y} \neq y \mid \hat{y}, t, q) = \min\{\eta(\hat{y}; t, q),\, 1 - \eta(\hat{y}; t, q)\}.$$

Let $h_2(p) := -p \log p - (1 - p) \log(1 - p)$ denote the binary entropy function. For any $p \in [0, 1]$, it holds that

$$h_2(p) \geq 2 \min\{p, 1 - p\}.$$

Applying this inequality with $p = \eta(\hat{y}; t, q)$ yields

$$2 \min\{\eta(\hat{y}; t, q), 1 - \eta(\hat{y}; t, q)\} \leq h_2(\eta(\hat{y}; t, q)).$$

Taking expectation over $\hat{y}$ conditional on $(t, q)$, we obtain

$$p_c(t, q) \leq \frac{1}{2} \, \mathbb{E}[h_2(\eta(\hat{y}; t, q)) \mid t, q].$$

By the definition of conditional entropy for a binary random variable,

$$\mathbb{E}[h_2(\eta(\hat{y}; t, q)) \mid t, q] = H(y \mid \hat{y}, t, q).$$

Therefore,

$$p_c(t, q) \leq \frac{1}{2} H(y \mid \hat{y}, t, q). \tag{11}$$

By definition of conditional mutual information,

$$I(\hat{y}; y \mid t, q) = H(y \mid t, q) - H(y \mid \hat{y}, t, q).$$

Rearranging gives

$$H(y \mid \hat{y}, t, q) = H(y \mid t, q) - I(\hat{y}; y \mid t, q).$$

Substituting into (11), we obtain

$$p_c(t, q) \leq \frac{1}{2}\Big(H(y \mid t, q) - I(\hat{y}; y \mid t, q)\Big).$$

Taking expectation over $(t, q)$ yields

$$p_{\text{error}} = \mathbb{E}_{t,q}\big[p_c(t, q)\big]$$
$$\leq \frac{1}{2}\Big(H(y \mid t, q) - I(\hat{y}; y \mid t, q)\Big).$$

Finally, since conditioning reduces entropy, $H(y \mid t, q) \leq H(y)$, we conclude that

$$p_{\text{error}} \leq \frac{1}{2}\Big(H(y) - I(\hat{y}; y \mid t, q)\Big).$$

This completes the proof. □

### A.9. Proof of proposition 3.2

**Proposition A.1** (Improved). *If $r_{\text{analytic}}(t_1) \geq r_{\text{analytic}}(t_2)$, and the conditional entropy given incorrect predictions does not increase too rapidly with $p(t, q)$, then*

$$H(y \mid \hat{y}, t_1, q) \leq H(y \mid \hat{y}, t_2, q).$$

*Proof.* Let

$$p_i := p(t_i, q) = \Pr(\hat{y} = y \mid t_i, q),$$

and define

$$C_i := H(y \mid \hat{y}, t_i, q, \hat{y} \neq y),$$

the expected conditional entropy when the prediction is incorrect.

From the law of total entropy,

$$H(y \mid \hat{y}, t_i, q) = p_i \cdot H(y \mid \hat{y} = y, t_i, q) + (1 - p_i) \cdot C_i.$$

Since $H(y \mid \hat{y} = y, t_i, q) = 0$ (the answer is known exactly when prediction is correct), we have

$$H(y \mid \hat{y}, t_i, q) = (1 - p_i)C_i. \tag{1}$$

Given $r_{\text{analytic}}(t_1) \geq r_{\text{analytic}}(t_2)$ implies $p_1 \geq p_2$, we have

$$1 - p_1 \leq 1 - p_2.$$

Let $C(p)$ denote the average conditional entropy given incorrect predictions as a function of $p = \Pr(\hat{y} = y \mid t, q)$. The total conditional entropy can be decomposed as

$$H(y \mid \hat{y}, t, q) = (1 - p)\, C(p),$$

since correct predictions contribute zero uncertainty and all remaining uncertainty arises from incorrect cases. Therefore, if $(1 - p)C(p)$ is non-increasing in $p$, increasing the success probability $p$ leads to a reduction in the overall conditional entropy.

This is equivalent to

$$\frac{d}{dp}\big[(1 - p)C(p)\big] = (1 - p)C'(p) - C(p) \leq 0,$$

or

$$C'(p) \leq \frac{C(p)}{1-p}.$$

This condition allows $C(p)$ to increase with $p$, but not too rapidly. Intuitively, a better plan (higher $p$) may still produce wrong answers, but their distribution over $y$ should not become much more dispersed.

From (A1) and $p_1 \geq p_2$, we have

$$(1 - p_1)C(p_1) \leq (1 - p_2)C(p_2).$$

Substituting into (1) gives

$$H(y \mid \hat{y}, t_1, q) \leq H(y \mid \hat{y}, t_2, q).$$

Finally, from the identity

$$I(y; \hat{y} \mid t, q) = H(y \mid t, q) - H(y \mid \hat{y}, t, q),$$

and noting that $H(y \mid t, q)$ is independent of the predictor $\hat{y}$, we obtain

$$I(y; \hat{y} \mid t_1, q) \geq I(y; \hat{y} \mid t_2, q). \qquad \square$$

### A.10. Experimental parameter setup

We conducted all experiments on eight H200 GPUs. In the supervised fine-tuning (SFT) stage, we trained Qwen2.5-7B-Instruct for 3 epochs. In the reinforcement learning (RL) stage, we adopted the GRPO algorithm, with a global batch size of 128 and a micro batch size of 4 per GPU. During rollout, the model generated 12 samples per step, including 4 analytic plans, each corresponding to 3 Chain-of-Thought (CoT) reasoning trajectories. For generation, we set temperature = 1.0 and top-p = 1.0, while for validation we used temperature = 0.6, top-p = 0.95. The number of RL training steps was configured as follows: LLaMA3.2-3B and Qwen2.5-7B-Instruct were trained for 350 steps, Qwen3-8B for 150 steps, and Qwen3-14B for 50 steps, with other hyperparameters kept the same across models. In addition, the learning rate (`lr`) was set to $1.0 \times 10^{-6}$, the weight decay (`weight_decay`) was $1.0 \times 10^{-2}$, the optimizer was `adamw` (choices: `adamw` or `adamw_bf16`), the learning-rate warmup ratio (`lr_warmup_ratio`) was 0. The hyperparameter $\beta$ is assigned to 0.5. For all Qwen3-8b, max token is 4.5k and for Qwen2.5-7B-Instrct , the max token is 3.5K.

### A.11. Test case of PTA-GRPO

## Generated Samples in Real Test cases of PTA-GRPO (Sample 1)

**Question:** Determine the interval of convergence for the series

$$1 + \frac{x}{2} + \frac{2!x^2}{3^2} + \frac{3!x^3}{4^3} + \dots$$

**<plan>**

1. Identify the general term of the series. 2. Apply the Ratio Test to find the radius of convergence. 3. Check the endpoints of the interval separately to determine convergence or divergence there. 4. State the final interval of convergence.

**<think>**

We start by writing down the general term of the series. The pattern is clear: the $n$-th term looks like

$$\frac{n!}{(n+1)^n}x^n.$$

To test for convergence, the Ratio Test is natural. Taking the ratio $a_{n+1}/a_n$, most factors cancel, and it reduces to

$$|x|\left(\frac{n+1}{n+2}\right)^{n+1}.$$

As $n \to \infty$, the power term tends to $e^{-1}$. So the limit is $|x|/e$. By the Ratio Test, convergence requires $|x|/e < 1$, which means $|x| < e$. That gives us the radius of convergence $R = e$.

Now check the endpoints. At $x = e$, the terms are $\frac{n!}{(n+1)^n}e^n$. Stirling's formula shows that $n! \sim (n/e)^n\sqrt{2\pi n}$, so the terms behave like $\sqrt{n}$ times a constant. They don't go to zero, so the series diverges. At $x = -e$, the alternating sign doesn't help because the size of the terms again grows like $\sqrt{n}$. They still don't go to zero, so the series diverges there as well.

Therefore, the series converges only strictly inside the interval, and the interval of convergence is

$$(-e, e).$$

**Answer:**

$$\boxed{(-e,\ e)}$$

## Generated Samples in Real Test cases of PTA-GRPO (Sample 2)

**Question:** What is the least possible value of the sum $|x - 1| + |x - 1.5| + |x - 2|$?

<plan>

1. Recognize that this is a minimization problem involving sums of absolute values. 2. Recall that for a sum of absolute values of linear functions, the minimum occurs at a median of the points. 3. Identify the points: 1, 1.5, and 2. 4. Determine the median of these three points. 5. Calculate the sum at the median point to find the minimum value.

<think>

We want to minimize

$$f(x) = |x - 1| + |x - 1.5| + |x - 2|.$$

General fact: For $|x - a_1| + \cdots + |x - a_n|$, the minimum occurs when $x$ is the median of the $a_i$.
Here, the three points are $1, 1.5, 2$. Ordered: $1 < 1.5 < 2$. So the median is $1.5$.
Evaluate at $x = 1.5$:

$$|1.5 - 1| + |1.5 - 1.5| + |1.5 - 2| = 0.5 + 0 + 0.5 = 1.$$

Check at neighbors:
* At $x = 1$: $0 + 0.5 + 1 = 1.5$. * At $x = 2$: $1 + 0.5 + 0 = 1.5$. * At $x = 1.25$: $0.25 + 0.25 + 0.75 = 1.25$. * At $x = 1.75$: $0.75 + 0.25 + 0.25 = 1.25$.
All larger than 1. Thus, the minimum occurs uniquely at $x = 1.5$, with value $1$.

**Answer:**

$$\boxed{1}$$

## Generated Samples in Real Test cases of PTA-GRPO (Sample 3)

**Question:** Find the range of $p$ for which the stability condition $|\xi| \leq 1$ holds, given that

$$|\xi|^2 = 1 - 4p^2(1 - p^2)\, s^4, \quad s = \sin\left(\tfrac{\omega}{2}\right) \in [0, 1].$$

**\<plan\>**

1. Understand the stability condition: $|\xi| \leq 1$. 2. Substitute the given expression for $|\xi|^2$. 3. The stability condition $|\xi| \leq 1$ is equivalent to $|\xi|^2 \leq 1$. 4. Since $|\xi|^2$ is expressed as $1 - 4p^2(1 - p^2)s^4$, ensure it is non-negative ($\geq 0$) to avoid complex values. 5. Analyze the sign of $p^2(1 - p^2)$. 6. Determine the condition on $p$ such that $|\xi|^2 \in [0, 1]$ for all $s \in [0, 1]$. 7. Conclude the range of $p$.

**\<think\>**

We have

$$|\xi|^2 = 1 - 4p^2(1 - p^2)s^4, \quad s \in [0, 1].$$

For stability, we need $|\xi| \leq 1$, i.e., $|\xi|^2 \leq 1$. That means

$$1 - 4p^2(1 - p^2)s^4 \leq 1.$$

This simplifies to

$$-4p^2(1 - p^2)s^4 \leq 0 \quad \Rightarrow \quad p^2(1 - p^2) \geq 0.$$

Since $p^2 \geq 0$, this requires $1 - p^2 \geq 0 \implies p^2 \leq 1$. Thus $|p| \leq 1$.
Additionally, we must ensure $|\xi|^2 \geq 0$. For $|p| \leq 1$, the term $4p^2(1 - p^2)s^4 \geq 0$, so $|\xi|^2 = 1 - (\text{nonnegative}) \in [0, 1]$. This guarantees both realness and the upper bound condition.
If $|p| > 1$, then $1 - p^2 < 0$, so $p^2(1 - p^2) < 0$. In this case the subtracted term becomes negative, effectively making $|\xi|^2 > 1$ for some $s$, which breaks stability.
Hence the stability condition holds exactly when

$$p \in [-1, 1].$$

**Answer:**

$$\boxed{[-1, 1]}$$

## Example of an LLM correcting an incorrect plan through self-reflection.

**Question:** Six points $A, B, C, D, E$, and $F$ lie in a straight line in that order. Suppose that $G$ is a point not on the line and that $AC = 26$, $BD = 22$, $CE = 31$, $DF = 33$, $AF = 73$, $CG = 40$, and $DG = 30$. Find the area of $\triangle BGE$.

**<plan>**

1. Assign coordinates to points A through F on the x-axis with A at 0. 2. Use the given segment lengths (AC, BD, CE, DF, AF) to set up equations and solve for all individual segment lengths AB, BC, CD, DE, EF. 3. Determine the coordinates of B, C, D, E, F. 4. Use distances CG and DG to locate point G by solving a system of equations from the distance formula. 5. Calculate area of triangle BGE using the shoelace formula with coordinates B, G, E. \\Wrong plan

**<think>**

Wait, I notice the plan is incorrect in its approach to finding point G. The plan says to use distances CG and DG to locate G, but this will only give us the x-coordinate of G, not both coordinates. Let me revise this. \\Revise it by LLM self-reflection

Looking more carefully at the problem, I need to find the area of triangle BGE. The points B, G, and E form a triangle where: - B and E are on the x-axis - G is somewhere in the plane

The area of triangle BGE can be calculated as ½ × —BE— × height from G to line BE. Since BE is horizontal (both B and E on x-axis), the height is simply —y-coordinate of G—.

So the revised approach should be: 1. Find coordinates of B and E 2. Find the y-coordinate of G 3. Area = $\frac{1}{2} \times |BE||y_G|$.

Let me implement this revised plan.

First, let's find all segment lengths as in the original plan: Let AB = a, BC = b, CD = c, DE = d, EF = e

From AC = 26: a + b = 26 From BD = 22: b + c = 22 From CE = 31: c + d = 31 From DF = 33: d + e = 33 From AF = 73: a + b + c + d + e = 73

From a + b = 26 and a + b + c + d + e = 73, we get c + d + e = 47 From c + d = 31 and c + d + e = 47, we get e = 16 From d + e = 33 and e = 16, we get d = 17 From c + d = 31 and d = 17, we get c = 14 From b + c = 22 and c = 14, we get b = 8 From a + b = 26 and b = 8, we get a = 18

So: AB = 18, BC = 8, CD = 14, DE = 17, EF = 16

Place A at (0,0), then: B = (18, 0) C = (26, 0) D = (40, 0) E = (57, 0) F = (73, 0)

Now, to find the y-coordinate of G: We know CG = 40 and DG = 30 Let G = (x,y)

CG² = (x-26)² + y² = 1600 DG² = (x-40)² + y² = 900

Subtract: (x-26)² - (x-40)² = 700 (x² - 52x + 676) - (x² - 80x + 1600) = 700 28x - 924 = 700 28x = 1624 x = 58

Now use (x-26)² + y² = 1600: (58-26)² + y² = 1600 32² + y² = 1600 1024 + y² = 1600 y² = 576 y = 24 (taking positive)

So G = (58, 24)

Now triangle BGE: B = (18, 0), E = (57, 0), G = (58, 24) BE = 57 - 18 = 39 Height = 24 (y-coordinate of G)

Area = ½ × 39 × 24 = 468

**Answer:**

| 468 |

**PTA-GRPO prompt**

```
{{ content | trim }}
Begin by outlining a high-level plan wrapped in <plan></plan> tags.
- This plan should capture only the major phases, strategic choices, and
conditional branches.
- Avoid low-level steps, calculations, or detailed reasoning here.  Next,
reason step by step within <think></think>.
- During reasoning, critically evaluate the initial plan.  If you find any
errors, inconsistencies, or improvements needed, revise your plan mentally
and continue reasoning based on the revised plan.
- Explicitly state if you are revising the plan and describe the changes.
- This is your detailed chain-of-thought:  work through assumptions,
intermediate steps, and logical derivations until the solution is reached.
Finally, provide the final answer enclosed within

\boxed{}
```

