# OpenReview forum: "Plan Then Action: High-Level Planning Guidance Reinforcement Learning  for LLM Reasoning"
_ICML.cc/2026/Conference — ICML 2026 regular_

### Official Review · Reviewer_Xp94 · 2026-03-05

**Soundness:** 3
**Presentation:** 3
**Significance:** 3
**Originality:** 3
**Overall Recommendation:** 5
**Confidence:** 3

**Summary:**

This work explores a major challenge in LLM reasoning: the absence of global planning in standard chain-
of-thought decoding, which operates as a token-level Markov Decision Process and is prone to local
myopia, redundant steps, and error propagation across long reasoning horizons. A central concept studied
by this article is the explicit integration of high-level analytic planning as both a structured supervision
signal and a first-class optimization target within a reinforcement learning framework. The proposed
method, PTA-GRPO, consists of two stages. In the first stage, a planning-structured SFT cold-start injects
analytic planning ability by training on data where ground-truth CoT reasoning has been summarized into
concise high-level plans. In the second stage, a guidance-aware RL phase extends GRPO with an analytic
plan reward defined as the empirical accuracy of CoT trajectories resampled under each candidate plan,
alongside the standard outcome reward and a format reward. The method is evaluated across five base
models and ten benchmarks spanning mathematical reasoning and multimodal understanding,
demonstrating consistent improvements over GRPO, DAPO, and several other RLVR baselines.

**Compliance With Llm Reviewing Policy:**

Affirmed.

**Key Questions For Authors:**

1. For Qwen3-8B, PTA-GRPO shows a small but statistically significant negative result on
MATH500 compared to GRPO. Could the authors offer a brief interpretation of this specific case,
for example, whether it reflects a format overhead cost on simpler problems that are already near-
saturated for this model?
2. Given that Table 7 shows self-generated plans perform only marginally below externally generated
ones, would the authors consider promoting this result more prominently in the main text as
evidence of the method's practical robustness?
3. Is there any plan to evaluate PTA-GRPO on non-mathematical reasoning tasks in a future version of
the paper, given the generality of the planning mechanism?

**Limitations:**

1. Diminishing improvements on stronger models
2. Limited evaluation scope across domains
3. Dependence on a very large model for plan generation

**Strengths And Weaknesses:**

Contributions:
1. Planning-Structured Reasoning Cold-Start: A dataset construction and SFT strategy that injects
explicit analytic planning ability into LLMs by training on plan-augmented CoT data, where plans are
generated by summarizing existing reasoning chains rather than requiring multi-turn interaction or external
planning modules.

2. Analytic Plan Reward in GRPO: A novel reward signal that measures plan quality as the empirical
accuracy of CoT trajectories resampled under each candidate plan, enabling direct optimization of
intermediate reasoning structure rather than relying solely on final answer correctness.

3. Theoretical Grounding via Mutual Information: A formal analysis connecting analytic plan reward
maximization to increases in the conditional mutual information between predicted and ground-truth
answers, with an upper bound on error probability derived from Fano's inequality in Theorem 3.1 and
validated empirically through HSIC estimation in Figure 3.

Strengths and Weaknesses:
S1: The paper addresses a genuinely important limitation of current RLVR methods. The observation that
outcome-based rewards like GRPO and DAPO provide no signal about the quality of intermediate
reasoning is well-motivated, and the proposed analytic plan reward is a principled and elegant response to
this gap. The motivating example in Figure 1 is clear and effectively illustrates how the absence of a
structured plan leads to a subtle but fatal reasoning error, making the problem immediately concrete and
accessible.

S2: The empirical evaluation is broad, thorough, and unusually rigorous for the field. The method is tested
across five model families, including LLaMA3.2-3B, Qwen2.5-7B-Instruct, Qwen3-8B, Qwen3-14B, and
Qwen2.5-7B-VL, and across ten benchmarks spanning mathematics and multimodal science. The inclusion
of statistical significance tests in Table 9 is commendable and relatively rare in the RLVR literature. The
results show that 13 out of 16 model-task pairs achieve significance at the 5% level or better, with
particularly strong and consistent gains on harder benchmarks like AIME24 and AIME25, where global
planning matters most.

S3: The ablation study in Table 3 is well designed and cleanly isolates the contribution of each component.
Removing the SFT stage causes by far the largest performance drop, from 58.46 to 46.33 in average score,
confirming that planning initialization is the most critical ingredient. Removing the analytic plan reward
causes a consistent and meaningful drop to 54.56, validating that the RL stage adds genuine value beyond
the SFT alone. The additional comparison in Table 4 between SFT with and without planning further
demonstrates that the planning supervision signal teaches transferable solution organization rather than
simply adding more training data.

S4: The theoretical analysis in Section 3.3 provides meaningful conceptual grounding. Theorem 3.1
establishes a clean connection between analytic plan reward maximization and reduction of the error
probability upper bound through conditional mutual information, and the empirical validation in Figure 3
shows that the plan reward and HSIC-estimated mutual information track together consistently across three
different model families. This alignment between theory and training dynamics adds credibility to the
proposed reward design in a way that most RL-for-reasoning papers do not attempt.

S5: The practical design of PTA-GRPO is thoughtful and efficient. The dual-use mechanism, where CoT
trajectories resampled to assess plan quality are also directly incorporated into the policy update, avoids any
additional sampling overhead compared to standard GRPO. The training time comparison in Appendix A.3
confirms that PTA-GRPO adds less than 3% overhead relative to GRPO on both Qwen2.5-7B-Instruct and
Qwen3-8B, making the method genuinely practical at scale.

W1: The improvements on the strongest models are more modest than on weaker ones, and two AMC23
results for Qwen3-8B and Qwen3-14B do not reach statistical significance. The paper reasonably attributes
this to a saturation effect as model capacity increases, and the gains on harder benchmarks like AIME24 and
AIME25 remain significant even for larger models. Nonetheless, a brief discussion of why AMC23
specifically shows diminishing returns, and whether this reflects task-level saturation rather than a
limitation of the method, would help readers better understand the scope of the contribution.

W2: The evaluation is concentrated in mathematical reasoning, with AIME24, AIME25, AMC23, and
MATH500 forming the core of the comparison in Table 1. The multimodal results in Table 5 are
encouraging and broaden the scope of the paper meaningfully, but a small additional experiment on a non-
mathematical planning-intensive task, such as multi-step question answering or code reasoning, would
further reinforce the generalization claims.

W3: The reliance on Qwen3-235B for plan generation in the SFT stage is noted but somewhat understated
in the main text. Table 7 in the appendix shows that self-generated plans achieve a competitive average of
57.37 versus 58.46 for externally generated plans, which is a reassuring result. It would strengthen the paper
to highlight this finding more prominently in the main text, since it demonstrates that the method does not
fundamentally require access to a very large external model and is therefore more broadly accessible than it
might initially appear.

---

> ### Author Rebuttal · Authors · 2026-03-31
>
> Dear Reviewer Xp94
>
> We thank the reviewer for the positive support for our paper. We hope that our response can address the reviewer’s concerns and further strengthen their confidence in this work.
>
> ### W1: Concerns on difference in gains
>
> We thank the reviewer for this thoughtful observation. The difference in improvement, we believe, is mainly due to the difference in benchmark difficulty.
>
> Specifically, AMC23 is relatively easier than AIME24 and AIME25, so stronger models already achieve high baseline performance under GRPO, leaving limited room for further improvement. In such near-saturated settings, the benefit of better planning is naturally smaller and less likely to reach statistical significance. In contrast, AIME24 and AIME25 rely more on global problem decomposition and multi-step coordination, so PTA-GRPO can still deliver clear and significant gains even for larger models.
>
> Therefore, we believe the smaller gains on AMC23 mainly suggest that the effect of planning is more pronounced on more complex tasks that truly require stronger global structure. We thank the reviewer again and will include this discussion in the revision.
>
> ### W2 & Q3: Generalization on other benchmark.
>
> Using Qwen2.5-7B-Instruct as the base model, we further evaluated the two methods on three different test tasks in MuSR [1], which is a multistep soft reasoning benchmark. Table X.1 shows that PTA-GRPO consistently outperforms GRPO across all three tasks, further suggesting that our method does not merely overfit to a specific setting, but instead exhibits better generalization ability.
>
> **Table X.1 Performance on the MuSR benchmark**
>
> | Method   | Murder Mysteries | Object Placements | Team Allocation |
> |----------|------------------|-------------------|-----------------|
> | Base     | 56.44            | 46.88             | 42.49           |
> | GRPO     | 57.57            | 47.39             | 43.29           |
> | PTA-GRPO | 59.33            | 50.10             | 44.70           |
>
> ### W3:Reduced Dependence on Large External Models
>
> We thank the reviewer for this helpful suggestion. We agree that this point deserves to be stated more explicitly in the main text. The plan signal introduced during the SFT stage essentially leverages the LLM’s ability to produce high-level summarization and abstraction based on the ground truth, and therefore does not fundamentally rely on support from a very large external model. A stronger external model mainly provides a convenient way to obtain slightly better supervision signals, while self-generated plans have also demonstrated strong effectiveness in practice. Following the reviewer’s suggestion, we will make this result more prominent in the main text to further emphasize the practical accessibility and broader applicability of our method.
>
> ### Q1
>
> We agree that, for Qwen3-8B, PTA-GRPO shows a small but statistically significant drop on MATH500 compared with GRPO. We believe this phenomenon is better interpreted as a localized saturation effect rather than a systematic weakness of the method. Based on our results, Qwen3-8B is already a strong baseline on MATH500, while PTA-GRPO still brings clear gains on more challenging benchmarks such as AIME24 and AIME25, and also improves the overall average performance over GRPO. This suggests that the main benefit of PTA-GRPO is more pronounced on harder tasks requiring longer-horizon reasoning.
> At the same time, we do not believe that introducing the explicit “plan-then-reason” on relatively simple tasks necessarily incurs additional overhead. In practice, the plan in our framework is usually very short, typically accounting for less than one-tenth of the CoT length. More importantly, when we measured the actual token usage on MATH500, we found that the average generation length of GRPO was 2270.4 tokens, whereas PTA-GRPO was in fact lower, at 2047.8 tokens. This suggests that explicit planning does not necessarily increase reasoning cost; instead, clearer high-level guidance can sometimes reduce redundant or drifting reasoning steps. We thank the reviewer for this helpful observation and will clarify this point more explicitly in the revision.
>
> ### Q2
>
> We agree that the result in Table 7 is important evidence of the practical robustness of our method. In particular, the fact that self-generated plans perform only marginally below externally generated ones suggests that PTA-GRPO does not rely heavily on plans provided by stronger external models, and can still remain effective when using plans generated by the model itself. We will therefore highlight this finding more prominently in the main text and further emphasize its practical implication in the revision.
>
> We hope that our response addresses your concerns, and we sincerely thank you again for your support.
>
> Best regards
>
> All authors of Submission30783
>
> [1] Musr: Testing the limits of chain-of-thought with multistep soft reasoning.

---

> > ### Author Rebuttal · Reviewer_Xp94 · 2026-04-01
> >
> > Thanks for the responses.

---

> > > ### Author Response · Authors · 2026-04-02
> > >
> > > Dear Reviewer Xp94
> > >
> > > Thank you for your valuable comments on our paper and for your strong support of our work. We sincerely appreciate your constructive feedback and suggestions, which have been very helpful in improving the quality and clarity of our paper. We have further revised and strengthened the manuscript based on your suggestions, and we will incorporate the corresponding analyses and experimental results into the main text and appendix. We also hope that these improvements will further strengthen your confidence in your assessment of our paper.
> > >
> > > **Thank you again for your time and effort. We have greatly benefited from this rebuttal process. We sincerely wish you all the best in your submissions and future research.**
> > >
> > > Best regards
> > >
> > > All authors of Submission30783

---

### Official Review · Reviewer_Zria · 2026-03-10

**Soundness:** 3
**Presentation:** 3
**Significance:** 3
**Originality:** 3
**Overall Recommendation:** 5
**Confidence:** 3

**Summary:**

This paper proposes PTA-GRPO, a two-stage training framework designed to enhance the reasoning capabilities of LLMs by explicitly introducing a high-level planning stage before detailed CoT generation. The framework consists of a Planning Structured Reasoning Cold-Start (PSR-CS) via supervised fine-tuning to inject planning abilities, followed by a Planning Structure-Guided Reinforcement Learning (PSG-RL) phase. The authors evaluate their method across multiple benchmarks, demonstrating performance improvements.

**Compliance With Llm Reviewing Policy:**

Affirmed.

**Final Justification:**

This paper proposes a simple yet effective method to guide the LLM reasoning by introducing the planning capability. Various experiments have validated the effectiveness from different perspectives. I initially had some concerns about the baseline and computation, but they have been well-addressed during the rebuttal period. Thus, I have improved my score to 5 and recommend acceptance for this paper.

**Key Questions For Authors:**

1. Can you provide a detailed comparison of the average number of generated tokens per problem for both GRPO and PTA-GRPO during evaluation? Was the total inference compute strictly controlled to ensure a fair comparison?
2. How does PTA-GRPO compare to a baseline where a strong base model is simply prompted to generate a plan before answering (e.g., zero-shot Plan-and-Solve prompting), especially when scaling test-time compute?

**Limitations:**

yes

**Strengths And Weaknesses:**

**Strengths:**
1. The approach is well-motivated by human cognitive processes, addressing the well-known issue of LLMs lacking global planning in token-level Markov Decision Processes.
2. The authors test their framework across various base models and benchmarks, demonstrating consistent gains over baseline RLVR methods.
3. The manuscript is well-written and easy to follow. The figures and tables are clear for presentation.


**Weaknesses:**
1. The introduction of the plan tag explicitly forces the model to generate additional tokens prior to the CoT. While the authors report comparable overall training times, it remains unclear whether the total token count during the RL rollouts and test-time evaluation was strictly controlled. If PTA-GRPO is generating significantly more tokens than the baselines, the performance gains might simply be a byproduct of increased computational resources rather than the architectural design.
2. The paper heavily emphasizes a complex RL pipeline to instill planning capabilities. However, modern models inherently possess deep thinking capacities, it is highly debatable whether this intensive training paradigm is necessary. A well-designed prompt guiding the model to plan and execute might achieve comparable test-time scaling benefits without the substantial overhead of RL training. The paper lacks a compelling baseline comparing PTA-GRPO against strong, prompt-driven test-time compute strategies on capable models.
3. Theorem 3.1 and its accompanying proof offer limited theoretical depth. The conclusion essentially states that maximizing the conditional mutual information $I(\hat{y}; y | t, q)$ minimizes the error bound. This is a standard information-theoretic truism and does not provide novel mathematical insights into the internal operating mechanisms of LLMs or how the planning tokens specifically restructure the representation space.

However, I don't think these shortcomings overshadow the thorough experimental verification in this paper.

After Rebuttal: I think my weakness 1&2 have been well-addressed. Weakness 3 remains as an open problem, but theoretical analysis is not the key objective of this paper, so it is okay. I will improve my score to 5.

---

> ### Author Rebuttal · Authors · 2026-03-31
>
> Dear Reviewer Zria
>
> We thank the reviewer for the positive support for our paper. We hope that our response can address the reviewer’s concerns and further strengthen their confidence in this work.
>
> ### W1 & Q1: Concerns on token length
>
> We thank the reviewer for raising this important concern. In fact, during both training and test-time inference, we use the same maximum generation length across all methods. Our method does not increase the token budget simply because it includes an explicit planning stage. Therefore, PTA-GRPO is not given extra generation budget compared with the baselines.
>
> Moreover, Figures 5 and 6(c) already show that, during training, our method actually uses fewer tokens than GRPO in practice. This indicates that the performance gains do not come from a larger computational budget or longer generations, but rather from the more efficient and better-structured reasoning enabled by the planning mechanism.
>
> To make this clearer, we will revise the paper to explicitly state that the maximum token budget is controlled identically across methods during both RL training and evaluation, and we will further highlight the empirical token-usage comparison shown in Figures 5 and 6(c).
>
> To provide a more comprehensive view of the test-time token costing, we further report the token consumption of the model during evaluation on different datasets, as shown below. The results show that PTA-GRPO does not gain an unfair advantage simply by generating longer reasoning sequences.
>
> **Table X.1 Comparison of response lengths, where Qwen2.5-7B-Instruct is used as the base model.**
>
> | Method   | MATH500 | AIME 24 | AIME 25 | AMC 23 |
> |----------|---------|---------|---------|--------|
> | GRPO     | 1295.9  | 3196.7  | 3329.2  | 1523.4 |
> | PTA-GRPO | 1097.3  | 2793.4  | 2914.0  | 1644.7 |
>
> ### W2 & Q2: Qualitative comparison analysis
>
> We thank the reviewer for this valuable baseline suggestion. In fact, Fig. 1(c) already provides preliminary evidence that, under the “plan-then-reason” framework, using self-generated plans from a relatively weaker model (e.g., Qwen2.5-7B-Instruct) to guide subsequent reasoning does not lead to stable performance. Although this strategy can improve performance in some cases, it can also hurt performance in others, and it is clearly less effective than using high-quality plans generated by a stronger model such as GPT-4o. This observation suggests that merely relying on prompting to elicit a “plan-first, then execute” behavior does not consistently produce effective global plans.
>
> Motivated by the reviewer’s suggestion, we further include a more direct baseline: under the same plan-reasoning inference format as PTA-GRPO, we remove all additional training and simply prompt the base model to first generate a plan and then perform the reasoning, denoted as Prompt-only in the table. We further compare it with both the SFT model and the fully trained model in Table X.2. The results further demonstrate that the training procedure is necessary.
>
> **Table X.2 Qualitative comparison analysis, where Qwen2.5-7B-Instruct is considered as base model.**
>
> | Method      | MATH500 | AIME 24 | AIME 25 | AMC 23 |
> |-------------|---------|---------|---------|--------|
> | Prompt-only | 64.24   | 11.73   | 5.44    | 53.95  |
> | SFT         | 80.40   | 25.25   | 20.33   | 63.75  |
> | PTA-GRPO    | 92.53   | 34.53   | 29.25   | 77.51  |
>
> ### W3: Furthermore theoretical analysis
>
> We agree that the purpose of Theorem 3.1 is not to explain the internal operating mechanisms of LLMs, nor to reveal how planning tokens restructure the representation space. Rather, this theorem is mainly intended to provide theoretical motivation for our reward design: under the guidance of the analytic plan, maximizing the conditional mutual information between the predicted answer and the ground-truth answer leads to a smaller upper bound on the prediction error, thereby explaining why the proposed plan reward can help steer the model toward reasoning outcomes that are closer to the ground truth.
>
> Therefore, our goal here is to provide theoretical intuition for why optimizing the analytic plan reward is beneficial for PTA-GRPO. We also agree that understanding how planning tokens affect the model’s internal representations is an important and worthwhile direction for deeper study. As such, we do indeed have a strong desire to propose a mechanistic theory regarding the internal structure of LLM. However, to the best of our knowledge, at present, we still lack sufficiently precise analytical tools to rigorously characterize this issue. We will further clarify this scope in the revision and highlight it as an important direction for future work.
>
> We hope that our response addresses your concerns, and we sincerely thank you again for your support.
>
> Best regards
>
> All authors of Submission30783

---

> > ### Author Rebuttal · Reviewer_Zria · 2026-04-01
> >
> > Thanks for your response. I think my concerns have been well-addressed, and I will improve my score to 5. Good luck!

---

> > > ### Author Response · Authors · 2026-04-02
> > >
> > > Dear Reviewer Zria
> > >
> > > Thank you for your valuable comments on our paper and for your strong support of our work. We also sincerely appreciate your support in raising our score. We sincerely appreciate your constructive feedback and suggestions, which have been very helpful in improving the quality and clarity of our paper. We have further revised and strengthened the manuscript based on your suggestions, and we will incorporate the corresponding analyses and experimental results into the main text and appendix. We also hope that these improvements will further strengthen your confidence in your assessment of our paper.
> > >
> > > **Thank you again for your time and effort. We have greatly benefited from this rebuttal process. We sincerely wish you all the best in your submissions and future research.**
> > >
> > > Best regards
> > >
> > > All authors of Submission30783

---

### Official Review · Reviewer_oJ5p · 2026-03-12

**Soundness:** 2
**Presentation:** 3
**Significance:** 1
**Originality:** 1
**Overall Recommendation:** 3
**Confidence:** 4

**Summary:**

This paper aims to improve the reasoning capacity of LLMs via a "plan-then-execute" paradigm: LLM learns to first output a high-level plan, then continues to generate detailed reasoning conditioned on this plan. To this end, the authors propose a two-stage training: starting with a warm-up stage with SFT to enable plan-then-execute reasoning pattern, PTA-GRPO then uses RL (GRPO) with multiple rewards (plan & reasoning) to incentivize the plan and reasoning simultaneously.

**Compliance With Llm Reviewing Policy:**

Affirmed.

**Final Justification:**

The rebuttal addresses many of my concerns, but I'm still not fully convinced on the issues of novelty and reward hacking (see the response below). For this reason, I will raise my score slightly, but I cannot support a higher rating.

**Key Questions For Authors:**

Apart from some issues mentioned in the weaknesses part, I also have some questions:
+ One important characteristic of the reasoning models is that they can change their reasoning paths as human beings with reflection, self-questioning, and etc. How does this ability integrate with the first-plan-then-execution paradigm? If the plan is erroneous, can the model learn to change the execution during generation?
+ For slow-thinking models, such as QWen3, will this plan-then-execution paradighm affects its slow-think pattern?
+ It seems that the designed reward does NOT necessarily incentivize a global plan. How to ensure it is a global plan?
+ For the experiments, the paper mentioned that 60K prompts are sampled from deepmath dataset. Is it randomized sampling? Please provide more details on the selection of the training dataset.
+ With 60K prompts and a batch size of 128, a 300-step training (Figures 3, 5, and 6) only consumes ~60% of the data. Why is not all the training data used?
+ For the Figure 5 and 6, why does the response length drop during the training?
+ The Figure 5 and 6 only show the training dynamics; could the authors also report how the test performance changes during the training?

**Limitations:**

yes

**Strengths And Weaknesses:**

## Strengths
+ Lacking overall planning during reasoning is indeed an important flaw of current LLMs
+ The presentation is good and easy to follow.

## Weaknesses
+ Though this paper aims to solve an important problem, I feel that the proposed idea is not novel enough; it just warms up with SFT, then incentivizes both the plan & reasoning with scalar rewards.
+ In general, planning is a high-level cognitive ability that requires much effort to develop, especially for the LLMs, as it is represented in short & concise tokens. Can such a challenging ability be well-initiated by a warm-up SFT? Can the authors provide more evidence that such planning ability is not just overfitting?
+ For the RL part,  a success-rate-like reward function is employed to quantify the planning quality. However, such a scalar reward cannot sufficiently determine the quality of planning. For example, a hint on the critical step also gives a high success rate, but it is NOT a global plan.
+ The overall reward simply sums all the rewards and can lead to reward advantage collapse, see GDPO[1]


[1] Liu et al., GDPO: Group reward-Decoupled Normalization Policy Optimization for Multi-reward RL Optimization.

---

> ### Author Rebuttal · Authors · 2026-03-31
>
> We have included all the experiments at the link https://anonymous.4open.science/r/icmlre-E5C1/README.md
>
> W1: novelty
>
> We respectfully disagree. Our method is not simply “SFT first, then RL with scalar rewards.” Its novelty has two aspects. First, the SFT stage uses plan-based supervision rather than flat CoT imitation. Standard reasoning SFT models $\pi_\theta(c \mid q)$, whereas ours learns a hierarchical target $\pi_\theta(t,c \mid q)$, where $t$ is a high-level plan. Thus, the model learns to plan before reasoning instead of only imitating reasoning traces. Table 4 shows that this plan signal improves reasoning accuracy. Second, our RL reward is fundamentally different from outcome-only RLVR. Rather than serving as a simple scalar signal on the final answer, it provides indirect supervision over the intermediate reasoning process by explicitly evaluating plan quality through multiple rollouts under the same plan. This mechanism is generally absent in prior outcome-only RLVR methods. In addition, Table 3 verifies its effectiveness. We further provide both theoretical and empirical analyses.
>
> W2:  Overfitting
>
> We do not claim warm-up SFT alone is sufficient to fully develop planning ability. In our framework, it provides a plan-conditioned initialization, and the subsequent RL stage further improves planning quality through rollout-based optimization. The two stages are complementary and both necessary. This design is also well motivated: prior top-tier work [1] shows that warm-up SFT can elicit higher-order reasoning behaviors such as reflection and self-correction, making it a natural way to introduce an explicit planning scaffold. Table 5 further shows strong cross-domain generalization on benchmarks far from the training distribution, suggesting that the gains come from improved planning rather than overfitting. We also evaluate on out-of-domain benchmarks such as MuSR [2]. Compared with GRPO, PTA-GRPO still improves on this benchmarks. Detailed results are in the supplementary experimental link.
>
> W3 and Q3:
>
> We believe our method can effectively evaluate plan quality. Assessing plan quality through downstream execution performance is a widely adopted practice in top-tier research [3]. In addition, we introduce an LLM-as-a-Judge evaluation [4], whose results show that better plans usually lead to higher correctness and thus higher scores, further supporting our reward design. Detailed results are provided in the link.
>
> Second, our design indeed avoids generating local hints. As shown in Appendix A.12, the prompt explicitly requires a high-level plan before execution, rather than scattered local hints. The ablation in Table 3 further shows that the plan reward improves reasoning performance.
>
> W4:
>
> Although GDPO suggests that GRPO may compress reward differences during advantage computation, this indicates a limitation rather than infeasibility. GRPO may not be optimal, but it remains effective in practice. Our method addresses a different and compatible direction, and our results already demonstrate its effectiveness. In future work, we will explore incorporating GDPO-style techniques into PTA-GRPO to further improve advantage computation.
>
> Q1:
>
> Even with an explicit planning stage, LLMs can still exhibit reflective behaviors during reasoning, such as revisiting assumptions or adjusting the reasoning path. Our framework does not force the model to stick to the initial plan in such cases. Moreover, our prompt explicitly encourages reflective reasoning, as shown in Appendix A.12. Therefore, the initial plan is not a rigid commitment, but a high-level guide for subsequent reasoning. Even if the initial plan is imperfect, the model can still adjust its execution during generation. A concrete example is provided in the fourth case of Appendix A.12.
>
> Q2:
>
> We empirically verify this on Qwen3-8B. In the slow-thinking setting, a limited token budget may cause the model to overthink, hurting final accuracy. PTA-GRPO mitigates this issue through its reasoning format and improves reasoning accuracy. Detailed analysis and case studies are provided in the link.
>
> Q4-5: We did not use the full dataset in order to align the data scale with prior work [1]. In addition, roughly 60% of the sampled data was already sufficient for our method to show clear gains over the baselines, which is enough to demonstrate its superiority and support our main claim.
>
> Q6: Response length drops during training because the SFT initialization uses OpenThoughts data, which tends to induce overly long outputs. During RL, the model is gradually adjusted toward a more appropriate response length.
>
> Q7: Test performance in the provided link.
>
> [1] SRPO: Enhancing Multimodal LLM Reasoning via Reflection-Aware Reinforcement Learning
> [2] MuSR: Testing the Limits of Chain-of-thought with Multistep Soft Reasoning
> [3] Learning to plan \& reason for evaluation with thinking-llm-as-a-judge
> [4] Agent-as-a-judge: Evaluate agents with agents

---

> > ### Author Rebuttal · Reviewer_oJ5p · 2026-04-03
> >
> > Thanks for the experiments and feedback. It helps address a part of my concerns, but some are still unsolved:
> >
> > [W1: novelty]
> >
> > I fully understand that this paper aims to enable the "plan-then-reasoning" capability. However, I feel the approach taken in this paper is quite common. For example, if someone wants to teach LLM to "search-then-reasoning", a similar approach will be adopted: first, warm up with some search-then-reasoning trajectories to let the LLM learn to call the search tool before reasoning. Then, similar RL training can be applied to further incentivize the model's capability[1, 2]. The only difference will be the reward design.
> >
> > [W3 and Q3: reward hacking issue]
> >
> > I'm also concerned that using a success-rate-like scalar reward function to quantify the planning quality. This may introduce certain "reward hacking": the model can learn to add some local hints in the plan part (say, "do check the corner cases when coding" or "do not forget to divide the dimension..."). These local hints can obviously improve the success rate, but are not global plans.   Could the authors clarify how they avoid such hacking issues in the current design?
> >
> >
> > [Q2: Experiment on slow-thinking model]
> >
> > In the experiment on Qwen-3-8B,  the maximal token number is set to 5K (Table 3 in the supplied link).  This can be far insufficient for challenging tasks like AIME and can hurt the baseline performance. Could the authors also report the comparison in a longer setting? say 32K.
> >
> > [1] REDSearcher: A Scalable and Cost-Efficient Framework for Long-Horizon Search Agents
> >
> > [2] WEBSEER: TRAINING DEEPER SEARCH AGENTS THROUGH REINFORCEMENT LEARNING WITH SELFREFLECTION

---

> > > ### Author Response · Authors · 2026-04-03
> > >
> > > We are glad that we have addressed part of your concerns, and we hope the following response can further engage in addressing the others.
> > >
> > > ## Concerns on Novelty
> > >
> > > We respectfully disagree with the statement that *“our RL is merely a change in the reward function.”* We have addressed this point in our previous response, and would like to further clarify it here.
> > >
> > > **The key distinction lies in the nature of analytic planning. Planning is an internal reasoning structure, rather than an externally grounded action such as search or tool use.** In search-then-reasoning settings [1] (that appears on arXiv after our paper submission) and [2] you mentioned, intermediate behaviors are tied to concrete tool calls and their outputs, making them relatively easy to supervise or evaluate. **In contrast, high-level plans are latent intermediate abstractions: their quality is not directly observable, nor independently verifiable. As a result, they cannot be effectively supervised under standard outcome-only RL.**
> > >
> > > Therefore, the core challenge is not simply to adopt a generic “X-then-reason” paradigm, but to make planning itself an objective that can be optimized under RL.
> > >
> > > Concretely, our method treats the plan as a first-class optimization target by sampling multiple candidates and estimating their quality via CoT rollout success rates. **As a result, the plan reward becomes a continuous signal, rather than a purely binary outcome-level supervision. This design provides a non-trivial credit assignment mechanism that propagates feedback from final outcomes back to high-level plans.** We further support our method with theoretical and empirical analyses from an information-theoretic perspective, providing insight into its effectiveness.
> > > This is fundamentally different from a new trajectory format with standard outcome-only RL.
> > >
> > > In summary, while SFT and RL are common method for improving model reasoning ability, our work differs from [1,2] in **motivation, objective, and technical route**, and is not merely a reward variation or a generic “X-then-reason” paradigm.
> > >
> > > ## Concerns on global plan
> > >
> > > In our prompt design (see Appendix), we explicitly constrain the `<plan>...</plan>` component to capture only high-level structure. Prompt clear requires: *it should include major phases, strategic choices, and conditional branches, while avoiding low-level steps, calculations, or detailed reasoning*. This design encourages the model to generate global guidance rather than local procedural details and hints.
> > >
> > > We do not exclude concise, high-level constraints in `<plan>...</plan>`. While brief and abstract, such guidance shapes downstream CoT across multiple steps, and is thus treated as actionable high-level guidance rather than local hints.
> > >
> > > From an optimization perspective, any guidance that consistently improves downstream reasoning is a valid component of a high-quality plan. Our goal is to learn guidance that is broadly effective across reasoning trajectories, rather than enforcing a strict linguistic definition of “global structure.”
> > >
> > > Following prior work [3,4], we use an *LLM-as-Judge* framework to evaluate plan quality (Table 2 in Link). We observe that plans yielding higher CoT accuracy receive higher quality scores, showing a strong positive correlation. This supports using downstream reasoning success as a practical optimization signal for plan quality.
> > >
> > >
> > >
> > > ## Concerns on slowing-thinking
> > >
> > >
> > >
> > > **Table 1 Comparison on Qwen3-8B in slow-thinking mode with a 32K token budget.**
> > >
> > > | Method                    | MATH500 ACC | MATH500 Len | AIME24 ACC | AIME24 Len | AIME25 ACC | AIME25 Len | AMC23 ACC | AMC23 Len |
> > > |---------------------------|------------|-------------|------------|------------|------------|------------|-----------|-----------|
> > > | Qwen3-8B-slow thinking    | 97.3       | 8255.7      | 78.57      | 13425.7    | 73.94      | 19247.5    | 95.57     | 8265.6    |
> > > | PTA reasoning format      | 98.2       | 6798.2      | 80.29      | 10608.5    | 74.62      | 13384.3    | 96.29     | 7739.7    |
> > >
> > > To address this concern, we conduct an experiment on Qwen3-8B with slow thinking and a 32K token budget, using accuracy (ACC) and response length (Len) as evaluation metrics. The results show that PTA-reasoning consistently outperforms the baseline across datasets in this long-context setting, achieving higher accuracy while significantly reducing token usage during inference.
> > >
> > > **Thank you for your thoughtful feedback, which has helped improve this paper.  We hope our clarifications address your concerns. We also wish you all the best in your submission and future research.**
> > >
> > >
> > > [1] REDSearcher: A Scalable and Cost-Efficient Framework for Long-Horizon Search Agents
> > >
> > > [2] WEBSEER: TRAINING DEEPER SEARCH AGENTS THROUGH REINFORCEMENT LEARNING WITH SELFREFLECTION
> > >
> > > [3] RECAP: REwriting Conversations for Intent Understanding in Agentic Planning EACL 2026
> > >
> > > [4] Learning to Plan & Reason for Evaluation with Thinking-LLM-as-a-Judge ICML 2025

---

### Official Review · Reviewer_ZKi7 · 2026-03-13

**Soundness:** 4
**Presentation:** 3
**Significance:** 3
**Originality:** 3
**Overall Recommendation:** 5
**Confidence:** 3

**Summary:**

This paper introduces PTA-GRPO, a framework to train LLMs to reason by first constructing high-level plans. The method consists of an initial SFT stage on high-level plans constructed by a teacher LLM from ground truth CoTs. Subsequently, an RL scheme based on GRPO is applied, in which the model first generates multiple high-level plans, and then multiple CoT trajectories per plan. Importantly, not only the final outcome is rewarded as in standard GRPO, but the method furthermore rewards the quality of the generated plans via their expected success probability under the rollout policy. The authors conduct a variety of experimental studies that demonstrate the positive effects on reasoning performance.

**Compliance With Llm Reviewing Policy:**

Affirmed.

**Final Justification:**

The authors' rebuttal has affirmed my positive assessment of the paper by adding new results and discussion. I maintained my positive assessment and think this is overall solid work.

**Key Questions For Authors:**

The proposed method naturally requires $m\cdot z$ rollouts, set to 12 in the experiments. It's appreciated that the baselines are evaluated with the same number of rollouts. However, how does the method perform under a more limited rollout budget?

**Limitations:**

yes

**Strengths And Weaknesses:**

**Strengths**

* The method is well-motivated. Planning out steps before attempting to solve a complex problem is natural and missing from most standard RLVR methods.
* The proposed pipeline of an SFT stage to induce general planning behaviour followed by an RL stage to enable improved reasoning makes sense.
* The idea of assigning of measuring the quality of a plan as the average performance under multiple rollouts conditioned on this plan is intuitive and rewards planning behaviour more directly than simply rewarding final outcomes.
* The authors conduct extensive experiments and ablation studies on different parts of the method.

**Weaknesses**

The paper is overall nicely structured, but there are some minor presentation issues and clarifying questions I have.

* The method separates the "planning" reward from the "outcome" reward and argues that both of these are crucial for performance. However, these are then simply combined into a single, overall reward for the entire generation. Have you considered instead applying the planning reward only to the tokens that actually form the plan, and the outcome reward only on the CoT tokens? A discussion of this in the paper would be appreciated.
* It should also be noted that the planning reward depends on the current policy, and is hence non-stationary. Technically this means we can no longer treat the proposed framework as an MDP.
* Given that the main motivation of the method is to enable models to learn better high-level planning, a (qualitative) comparison of plans produced with prompting alone, after SFT, and after RL would be appreciated.
* The results overall are strong, and the statistical analysis in the appendix is appreciated. However, can you provide confidence intervals for the results in the main paper?
* There are some minor inconsistencies:
	* Eq. (1) should be argmin
	* I suppose the softmax in Eq. (2) is over analytic plans, but this could be made clearer
	* Appendix title is "Apprndix"

---

> ### Author Rebuttal · Authors · 2026-03-31
>
> Dear Reviewer ZKi7
>
> We thank the reviewer for the positive support for our paper. We hope that our response can address the reviewer’s concerns and further strengthen their confidence in this work.
>
> ### W1: Fine-Grained Reward Assignment
>
> Following the reviewer’s suggestion, we conducted an additional experiment in which we separately computed the advantages for the plan reward and the task reward, and applied them to the plan part and the corresponding CoT part of the model response, respectively, named as PTA-GRPO-S. The results in Table X.1 show that PTA-GRPO-S performs worse than PTA-GRPO. We hypothesize that planning and reasoning are tightly coupled, so applying a unified reward over the full trajectory may provide a more coherent training signal.
>
> **Table X.1 Comparison between PTA-GRPO and PTA-GRPO-S, where Qwen2.5-7B-Instruct is used as the base model.**
>
> | Method     | MATH500 | AIME 24 | AIME 25 | AMC 23 |
> |------------|--------:|--------:|--------:|-------:|
> | PTA-GRPO-S | 86.62   | 29.22   | 25.57   | 72.50  |
> | PTA-GRPO   | 92.53   | 34.53   | 29.25   | 77.51  |
>
> ### W2: Statement of MDP
>
> We agree that the proposed planning reward is policy-dependent, since it is computed based on rollouts generated by the current policy. Therefore, strictly speaking, the resulting optimization problem should not be viewed as a standard stationary MDP with a fixed reward function.
> However, this does not affect the validity of PTA-GRPO as an on-policy optimization procedure. At each training step, both the reward computation and the policy update are performed based on trajectories sampled from the current policy. In this sense, the non-stationarity arises from the rollout-dependent reward construction, rather than from any inconsistency between the optimization objective and the sampled data.
> We appreciate the reviewer for highlighting this subtle but important issue. In the revised paper, we will further clarify that our framework is more accurately characterized as an on-policy optimization procedure with policy-dependent reward construction, rather than a standard stationary MDP with a fixed reward function.
>
> ### W3: Qualitative Comparison
>
> Following the reviewer’s suggestion, we further added this experiment on the Qwen2.5-7B model to compare plans generated by prompting alone, after SFT, and after RL. The results are in Table X.2
>
> **Table X.2 Qualitative comparison analysis, where Qwen2.5-7B-Instruct is used as the base model.**
> | Method      | MATH500 | AIME 24 | AIME 25 | AMC 23 |
> |-------------|--------:|--------:|--------:|-------:|
> | Prompt-only | 64.24   | 11.73   | 5.44    | 53.95  |
> | SFT         | 80.40   | 25.25   | 20.33   | 63.75  |
> | PTA-GRPO    | 92.53   | 34.53   | 29.25   | 77.51  |
>
> ### W4: confidence intervals
>
> We will therefore add confidence intervals for the main results in the revised version. The corresponding confidence intervals are shown in Table X.3
>
> **Table X.3 Confidence Intervals of GRPO and PTA-GRPO, where Qwen2.5-7B-Instruct is used as the Base Model**
>
> | Task    | GRPO 95% CI   | PTA-GRPO 95% CI |
> |---------|---------------|-----------------|
> | MATH500 | [88.36, 90.98] | [91.22, 92.84] |
> | AIME24  | [29.41, 30.89] | [33.90, 35.16] |
> | AIME25  | [24.52, 25.88] | [28.54, 30.96] |
> | AMC23   | [72.48, 74.94] | [76.89, 78.53] |
>
> We will add this in our revision.
>
> ### W5: Minor typos
>
> We thank the reviewer for pointing out these minor inconsistencies. We will correct them in the revised version.
>
> ### Q1:Future experiment of rollout number
>
> We have already included a sensitivity analysis on different rollout budgets in Figure 7 and Table 8, where we report the performance under different choices of
> $m$ and
> $z$. The corresponding accuracy trends can be directly observed.
>
> To comprehensively evaluate the performance of PTA-GRPO under a limited rollout budget, we set $m=2$ and $z=3$, which corresponds to a group size of 6 in GRPO. The comparative results are shown in Table X.4 .
>
> **Table X.4 Comparison of GRPO and PTA-GRPO under Limited Rollout Budgets**
>
> | Method   | MATH500 | AIME 24 | AIME 25 | AMC 23 |
> |----------|---------|---------|---------|--------|
> | GRPO     | 84.75   | 28.22   | 24.57   | 68.38  |
> | PTA-GRPO | 87.29   | 31.72   | 26.94   | 71.95  |
>
> We hope that our response addresses your concerns, and we sincerely thank you again for your support.
>
> Best regards
>
> All authors of Submission30783

---

> > ### Author Rebuttal · Reviewer_ZKi7 · 2026-04-01
> >
> > Many thanks for the response.
> >
> > > Table X.2
> >
> > For the final version of the paper, it could also be nice to demonstrate a few examples of the differences in plans produced after the different training stages, but this is just a suggestion.
> >
> > Overall, I think this paper was already very solid and the new experiments/presentation will improve it further. Good luck with the submission!

---

> > > ### Author Response · Authors · 2026-04-02
> > >
> > > Dear Reviewer ZKi7
> > >
> > > Thank you for your valuable comments on our paper and for your strong support of our work. We sincerely appreciate your constructive feedback and suggestions, which have been very helpful in improving the quality and clarity of our paper. We have further revised and strengthened the manuscript based on your suggestions, and we will incorporate the corresponding analyses and experimental results into the main text and appendix. We also hope that these improvements will further strengthen your confidence in your assessment of our paper.
> > >
> > > **Thank you again for your time and effort. We have greatly benefited from this rebuttal process. We sincerely wish you all the best in your submissions and future research.**
> > >
> > > Best regards
> > >
> > > All authors of Submission30783

---

### Decision · Program_Chairs · 2026-04-30

**Decision:**

Accept (regular)

**Comment:**

This submission proposes PTA-GRPO. It is a two-stage framework for improving LLM reasoning. Its idea is to explicitly introduce high-level planning before detailed chain-of-thought generation. The proposed method uses planning-structured SFT to teach a plan-then-reason format, and then applies a RL procedure that optimizes the quality of the generated plan.

There is consensus among the reviews with three acceptance recommendation and one weak reject (after the rebuttal). The rebuttal clarified several technical questions raised by reviewers. The author(s) added new experiments on rollout budget, token usage, prompt-only baselines, and ood generalization, thus resolving most concerns.

The major strengths of this submission (from reviewers):
- 1 Well-motivated problem and the overall two-stage design was considered coherent and technically sound.
- 2 Strong empirical validation.
- 3 Practical effectiveness of explicit plan optimization.

The weaknesses raised by reviewers:
- 1.	Questions about novelty relative to existing plan-/tool-assisted reasoning pipelines. Although the rebuttal clarified the distinction between optimizing latent analytic plans and supervising externally grounded actions, this reviewer still viewed the originality as somewhat limited (oJ5p).
- 2.	ambiguity in the concept of plan. The authors argued that the prompt format explicitly constrains the plan to high-level guidance and provided additional judge-based evidence, but this issue was not viewed as fully settled by all reviewers (oJ5p).

Overall, the submission receives positive ratings, and makes a well-evaluated contribution on LLM reasoning.. While there is some disagreement about the degree of novelty and about how strongly the current reward design guarantees genuinely global plans, the overall reviewer consensus is positive. Three reviewers support acceptance and viewed their main concerns as fully or largely resolved after rebuttal, with particular acknowledgement for the empirical breadth, clarity of presentation, and practical effectiveness of the method. Therefore, it is relatively clear to give an acceptance recommendation for this submission.